# Causal involvement of dorsomedial prefrontal cortex in learning the predictability of observable actions

Pyungwon Kang [1] ✉, Marius Moisa [1], Björn Lindström [2], Alexander Soutschek [3], Christian C. Ruff [1] & Philippe N. Tobler [1,4]

Social learning is well established across species. While recent neuroimaging studies show that dorsomedial prefrontal cortex (DMPFC/preSMA) activation correlates with observational learning signals, the precise computations that are implemented by DMPFC/preSMA have remained unclear. To identify whether DMPFC/preSMA supports learning from observed outcomes or observed actions, or possibly encodes even a higher order factor (such as the reliability of the demonstrator), we downregulate DMPFC/preSMA excitability with continuous theta burst stimulation (cTBS) and assess different forms of observational learning. Relative to a vertex-cTBS control condition, DMPFC/preSMA downregulation decreases performance during action-based learning but has no effect on outcome-based learning. Computational modeling reveals that DMPFC/preSMA cTBS disrupts learning the predictability, a proxy of reliability, of the demonstrator and modulates the rate of learning from observed actions. Thus, our results suggest that the DMPFC is causally involved in observational action learning, mainly by adjusting the speed of learning about the predictability of the demonstrator.

Learning from the actions of others is crucial for agents to acquire information without having to directly experience the associated outcomes[1]. In accordance with its efficiency and adaptive value, observational learning is widely used across species[2–5]. For example, by observing conspecifics, humans learn novel actions, language, and behavioral strategies already from an early age[3,6]. Observational learning also helps chimpanzees to use novel tools[2] and mice to acquire stimulus-reinforcer associations[7].

A growing body of work has examined the neural basis of observational learning. Human fMRI studies have highlighted the involvement of posterior dorsomedial prefrontal cortex, just in front of the pre-supplementary motor area, in observational learning[8–12]. The area is often referred to as posterior medial frontal cortex. However, we designated it as dorsomedial prefrontal cortex/pre-supplementary motor area (DMPFC/preSMA) to be consistent with the study[10] from

which we took the coordinates for stimulation, but we also specify its posterior location within the DMPFC. Previous research has shown that DMPFC/preSMA activity is correlated with key processes in observational learning, such as the difference (prediction error) between observed and simulated actions[10] and the difference between simulated and observed outcomes[8]. Moreover, DMPFC/preSMA activity relates to updates in the emulation of the demonstrator (i.e. an observed person from whom one learns)[11]. A non-human primate study revealed that some neurons in the medial frontal gyrus – a homologue to human DMPFC - process specifically the observed behavioral errors of other monkeys[13]. All these studies suggest that DMPFC/preSMA plays a central role in the network of brain regions contributing to observational learning. However, the evidence for the specific functional contributions of DMPFC/preSMA to observational learning is rather multi-faceted, with different studies highlighting correlations

[1]Zurich Center for Neuroeconomics, Department of Economics, University of Zurich, Zurich, Switzerland. [2]Department of Clinical Neuroscience, Division for Psychology, Karolinska Institute, Stockholm, Sweden. [3]Ludwig Maximilian University Munich, Department for Psychology, Munich, Germany. [4]Neuroscience Center Zurich, ETH Zurich and University of Zurich, Zurich, Switzerland. ✉e-mail: pyungwon.kang@gmail.com

with conceptually different functions, such as action-prediction, outcome-prediction, or emulation of the demonstrator[9–12].

Here, we aimed to clarify the causal contribution of DMPFC/preSMA to observational learning, by combining continuous theta burst stimulation (cTBS) with a computational approach that allowed us to directly examine the predictions of three main proposals in the literature. First, given that DMPFC/preSMA activity correlated with simulated outcome- and action prediction errors in previous fMRI studies, DMPFC/preSMA cTBS may directly disrupt learning from (1) observed outcomes and/or (2) action prediction errors. Alternatively, DMPFC/preSMA may be involved in (3) a higher-level process, such as monitoring the performance of the observed person and using this information to modulate one's own observational learning process[14]. According to this latter view, the DMPFC/preSMA would be more crucial for evaluating whether to rely on a particular demonstrator during learning, rather than for directly learning from observed outcomes and/or actions. This perspective comes from reports that demonstrator performance has considerable influence on observational learning performance[15–17] and that children from the age of 3 track information about the demonstrator, such as their reliability and recent accuracy[18,19]. However, so far these alternative views on how exactly DMPFC/preSMA contributes to observational learning have not been assessed with causal neuroscience methods.

Outside of the observational learning literature, many studies have shown that the DMPFC/preSMA is involved in performance monitoring of self and others, and in subsequent performance adjustment across different types of tasks[14,20]. It is therefore plausible that the DMPFC/preSMA may be involved in adjusting observational learning based on the performance of the demonstrator. Specifically, we hypothesized that learning about the predictability of demonstrator actions would be affected by DMPFC/preSMA cTBS. Predictability is an observable indirect index of demonstrator performance regardless of the type of available information (outcome or action) and seems to be critical in determining the use of social information[19].

To distinguish among these competing accounts, we modified a previously established[8,9,15] observational learning task (Fig. 1A) and used a within-subject design with offline cTBS of DMPFC/preSMA (Fig. 1B) versus vertex. Participants learned stimulus-reward associations by observing a demonstrator choose from a two-armed bandit. In different conditions, participants viewed either only the actions (Action-Only) or also the outcomes (Action-Outcome) of a demonstrator in the observation phase and then had the opportunity to choose for themselves (and obtain the resulting rewards) in the decision phase. Crucially, the participants could not see the outcomes of their own actions in the decision phase but were informed of their payoff only after the experiment, so that they had to observe the demonstrator to learn about the most promising actions.

In conditions dissociating outcome-based from action-based learning, participants learned from either a superbly performing demonstrator (≥ 70% correct) or a badly (50% correct, i.e., randomly) performing demonstrator. Observing the actions of the bad demonstrator could not increase the performance of the participant; accordingly, participants should rely less on the observed actions and more on the observed outcomes of the bad demonstrator. On the other hand, it is beneficial to rely on, and learn from, the observed actions of the superb demonstrator even in the absence of outcome information[9]. Accordingly, the two demonstrators differed not only in whether their actions or their outcomes were more informative, but also in their predictability. This allowed us to examine which component of observational learning (i.e., outcome, action, or predictability) was affected by DMPFC/preSMA cTBS, through behavioral analyses and computational models. Furthermore, participants made both choices for themselves and predicted the choices of the demonstrator. The separation of what participants learned about the action tendencies of the demonstrator (given by the predictions of the participants) from how they used this information (given by their own choices) allowed us to specify the contributions of DMPFC/preSMA to observational learning without unspecific decision-making confounds.

Here, we show that downregulation of DMPFC/preSMA with cTBS decreases learning from, as well as accuracy in the predictions of, the actions of the demonstrator compared to vertex cTBS, reflecting reduced learning from observed actions. Furthermore, a computational approach reveals that the observed decline in action-based learning arises from disrupted learning about the predictability of the demonstrator. In the best-fitting model, the learned predictability of demonstrator actions dynamically modulates action-based learning and serves as a proxy for demonstrator reliability. Our findings thereby specify the computational role of the DMPFC/preSMA in observational learning.

## Results

To assess performance of our participants, we conducted a generalized linear mixed-model analysis with *choice for self* (1 = high reward probability option, 0 = low reward probability option) as dependent variable. First, for exploratory analyses, we regressed the probability of choosing the higher reward probability option on predictors for stimulation site (vertex, DMPFC/preSMA), observed information (Action-Outcome, Action-Only), demonstrator quality (superb, bad), as well as the interactions between these three variables (see Methods for details). As expected, participants performed better when they observed the superb demonstrator than when they observed the bad demonstrator, $b = -0.487$, $z = -2.219$, $p = 0.026$ (all effects remained qualitatively the same and significant or non-significant when controlling for stimulation order), 95% confidence interval (95% CI) = [−0.89 −0.07], Bayes factor$_{10}$ = 5.08. Moreover, demonstrator quality interacted with observed information, $b = -0.524$, $z = -2.123$, $p = 0.033$, 95% CI = [−0.99 −0.04], Bayes factor$_{10}$ = 3.35. Inspection of the data revealed (Fig. 2) that performance was more strongly enhanced by observing the superb demonstrator rather than the bad demonstrator in the Action-Only condition. No other significant two-way or three-way interactions were found.

Next, we focused on planned pairwise comparisons between DMPFC/preSMA downregulation and vertex stimulation, to investigate which type of learning was mostly influenced by brain stimulation. When participants could observe only actions but not outcomes, the DMPFC/preSMA vs. vertex cTBS comparison for the superb demonstrator reveals how a functioning DMPFC/preSMA contributes to observational action learning (without any outcome information), while the corresponding comparison for the bad demonstrator is less informative (because observing unreliable actions only carries little information). By contrast, when participants observe both actions and outcomes, the DMPFC/preSMA vs vertex cTBS comparison for the bad demonstrator in particular demonstrates the role of DMPFC/preSMA in outcome-based learning.

Thus, we performed planned pairwise comparisons to test the effect of brain stimulation in the Action-Only condition with the superb demonstrator, which investigates stimulation effects on action-based learning, and in the Action-Outcome condition with the bad demonstrator, which assesses stimulation effects on outcome-based learning. DMPFC/preSMA downregulation compared to control (vertex) stimulation decreased performance in the Action-Only condition with the superb demonstrator, $b = 0.341$, $z = 2.536$, $p = 0.011$, 95% CI = [0.08 0.61], Bayes factor$_{10}$ = 7.26. In contrast, Action-Outcome learning from the bad demonstrator remained unimpaired, $b = 0.016$, $z = 0.078$, $p = 0.937$, 95% CI = [−0.39 0.43], Bayes factor$_{10}$ = 0.02, (Fig. 2A). Thus, DMPFC/preSMA cTBS appeared to decrease action-based observational learning rather selectively. We also examined if DMPFC/preSMA downregulation affected Action-Outcome learning with superb demonstrators and found that DMPFC/preSMA downregulation had

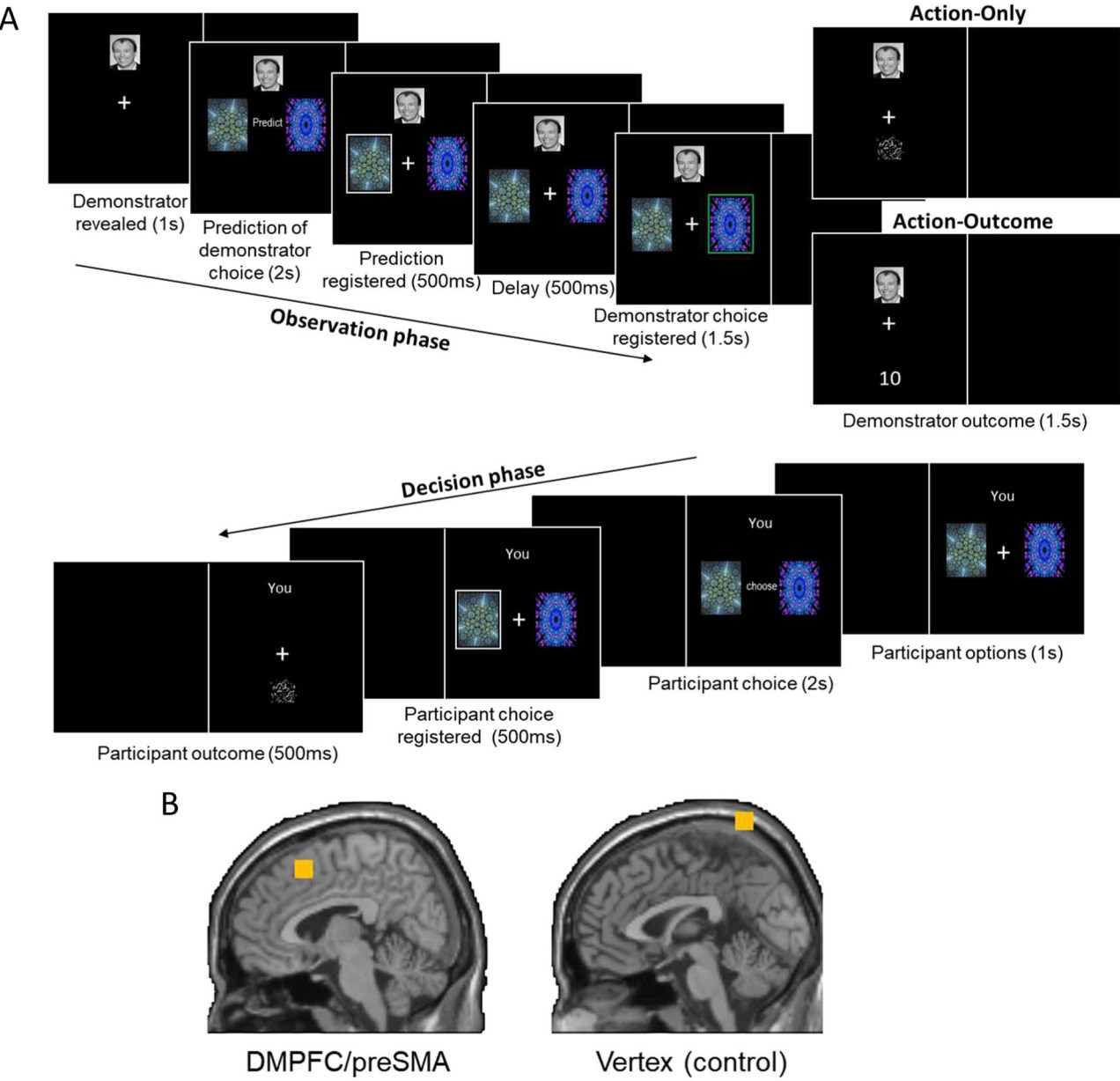

**Fig. 1 | Trial structure of the observational learning task and stimulation sites.**
**A** The observation phase started with a photo of the demonstrator, followed by a pair of fractal images. Participants then predicted which image the demonstrator would choose. After a delay, the actual choice of the demonstrator was revealed. Next, the outcome of the demonstrator appeared, unscrambled in the Action-Outcome condition or scrambled in the Action-Only condition. In the decision phase, the same pair of fractal images was displayed. The participant chose one of the images. The outcome of the participants was scrambled, forcing them to learn from observed information in the observational learning conditions. **B** Brain stimulation sites in DMPFC/preSMA (x = 6, y = 14, z = 52) and vertex.

no significant effect, b = 0.130, z = 0.523, p = 0.6.6, 95% CI = [−0.36 0.62], Bayes factor$_{10}$ = 0.07. Thus, it is possible that outcome-based learning compensated for the decrease in action-based learning. Lastly, no statistically significant difference was found in Action-Only learning from bad demonstrators, b = 0.153, z = 0.760, p = 0.447, 95% CI = [−0.24 0.54], Bayes factor$_{10}$ = 0.05.

In order to test whether the stimulation effects were limited to observational learning, we examined an individual learning condition, where observation was uninformative (see Methods). Purely individual learning performance was also unaffected by DMPFC/preSMA cTBS, b = −0.015, z = −0.407, p = 0.686, 95% CI = [−0.06 0.09], Bayes factor$_{10}$ = 0.03. Moreover, there was no effect of sex, b = 0.102, z = −0.071, p = 0.502, 95% CI = [−0.00 0.19]. Taken together, these results suggest that DMPFC/preSMA plays a central role mainly for action-based observational learning.

For prediction of demonstrator actions, as expected, a generalized linear mixed model revealed that participants predicted the choices of the superb demonstrator more accurately than those of the bad demonstrator, b = −1.477, z = −9.784, p < 0.001, 95% CI = [−1.77 −1.18], Bayes factor$_{10}$ = 1.94*10$^8$. Observing both actions and outcomes showed no statistically significant difference in prediction performance compared to observing only actions, b = −0.25, z = −1.73, p = 0.082, Bayes factor10 = 0.58. Next, we investigated the effects of DMPFC/preSMA downregulation on *prediction of demonstrator actions* in the observation phase. Given that DMPFC/preSMA cTBS decreased the learning performance, we examined whether cTBS reduced the ability to keep track of the decisions of the demonstrators, or only the use of tracked information for own decisions. If DMPFC/preSMA downregulation disrupted the ability to track observed behavior, then prediction performance in the

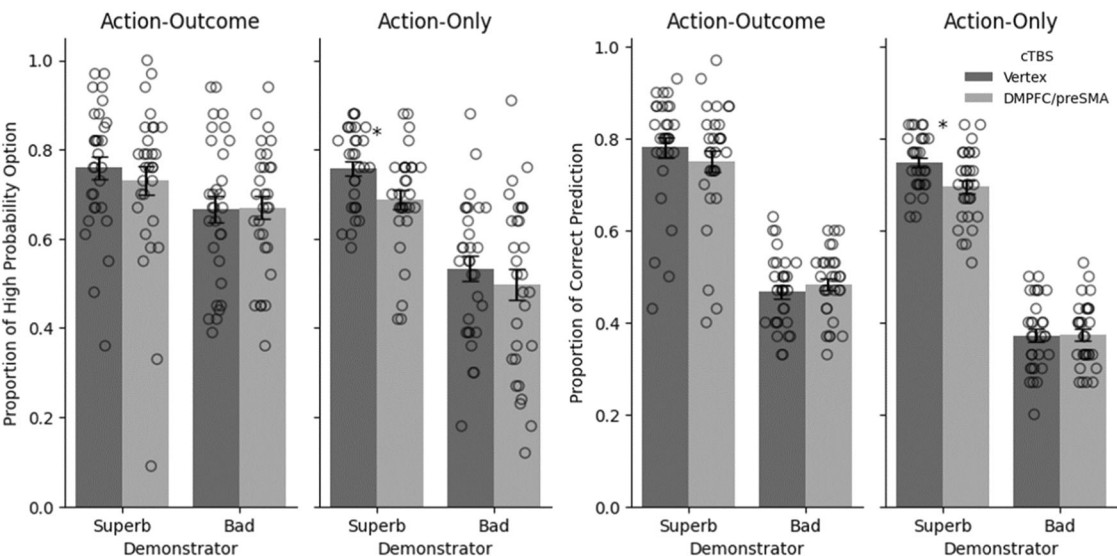

**Fig. 2 | Choice for self and prediction of demonstrator actions performance after cTBS over DMPFC/preSMA and vertex.** A Compared to vertex stimulation, downregulation of DMPFC/preSMA significantly reduced the propensity of choosing the higher reward probability option (mean ± SEM) after observing only the actions of the superb demonstrator ($n = 30$), $p = 0.011$, two-tailed. B Compared to vertex stimulation, downregulation of DMPFC/preSMA significantly reduced the proportion of correct predictions (mean ± SEM) of actions performed by the superb demonstrator in the Action-Only condition ($n = 30$), $p = 0.018$, two-tailed. * indicates $p < 0.05$. Error bars indicate the standard error (SE).

observation phase should decrease together with choice performance in the decision phase. In contrast, if DMPFC/preSMA downregulation reduced the ability of participants to use acquired information for their own choices, then prediction performance should remain unaffected in the Action-Only condition with the superb demonstrator (note that predictions are naturally close to chance when observing the bad demonstrator). We again examined this issue with planned pairwise comparisons. These revealed that compared to vertex stimulation, DMPFC/preSMA downregulation resulted in poorer prediction performance with the superb demonstrator in the Action-Only condition, $b = 0.264$, $z = 2.368$, $p = 0.018$, 95% CI = [0.03 0.49], Bayes factor$_{10}$ = 0.78, but neither in the Action-Outcome condition, $b = 0.169$, $z = 1.373$, $p = 0.169$, 95% CI = [−0.06 0.40], Bayes factor$_{10}$ = 0.08, nor in the Action-Outcome learning condition with the bad demonstrator, $b = 0.067$, $z = −0.681$, $p = 0.496$, 95% CI = [−0.10 0.24], Bayes factor$_{10}$ = 0.03; Fig. 2B). Moreover, there was no effect of sex, $b = 0.021$, $z = 0.402$, $p = 0.692$, 95% CI = [−0.21 0.25]. Thus, the stimulated DMPFC region appears to be important for keeping track of demonstrator actions, not just for using this information to guide own behavior.

Given that choice and prediction performance in the Action-Only condition were impaired by DMPFC/preSMA downregulation, one may ask if DMPFC/preSMA downregulation affected the ability of our participants to imitate (i.e., to select exactly the same options as the demonstrator in each trial) rather than the ability to learn from observed actions. We differentiated between action-based learning and imitation by examining whether the accumulation of knowledge across multiple trials was necessary or not. A generalized linear mixed model analyzing imitation behavior (see Methods) revealed no statistically significant difference between vertex and DMPFC/preSMA downregulation in either condition (Figure S3). Moreover, planned pairwise comparisons for stimulation site found no statistically significant difference between vertex and DMPFC/preSMA downregulation. In addition, we examined a model with imitation behavior instead of action based learning; it fit the data worse than most other models (SI). Thus, we find little support for stimulation effects on imitation. This means that the DMPFC/preSMA cTBS effect in the Action-Only condition with the superb demonstrator did not result

from impaired imitation, but from impaired observational action learning.

## Computational model for choice for self and prediction of demonstrator actions

The model-independent analyses demonstrated that DMPFC/preSMA cTBS decreased both observational learning (*choice for self*) and the *prediction of demonstrator actions*, specifically when participants observed a superb demonstrator. While this is consistent with a selective role of DMPFC/preSMA in observational action learning, it does not answer which specific learning process was affected by cTBS. Pinpointing the specific latent process implemented by the DMPFC/preSMA during observational learning with computational modeling requires systematic model variation and comparison within a common model structure, in keeping with the notion that computational modeling needs to be tailored to the diverse purposes it can serve[21]. Accordingly, rather than comparing widely different and novel models, what is needed is to build upon, and systematically advance, previous modeling approaches[8,9] that were based on a closely related behavioral paradigm.

To understand the exact role of the DMPFC/preSMA for observational learning, we examined two groups of computational models (Methods section) that formally model and distinguish 1) learning from observed outcomes, 2) learning from observed actions, and 3) learning about the predictability of the demonstrator (Fig. 3A; Figure S6 with equations). The first group of these models tested direct effects of DMPFC/preSMA cTBS on outcome- and/or action-based learning while the second group tested whether DMPFC/preSMA downregulation affected learning about the predictability of the demonstrator when observing outcomes and/or actions. Specifically, Models 1-3 of the first group tested whether DMPFC/preSMA cTBS altered outcome-based learning (Model 1, Outcome learning model), action-based learning (Model 2, Action learning model), or both outcome- and action-based learning (Model 3, Action-Outcome model) (see (Fig. 3A)).

Rather than directly learning from the actions or outcomes of the demonstrator, the models in the second group (Models 4-6) tested whether learning about the predictability of the demonstrators was

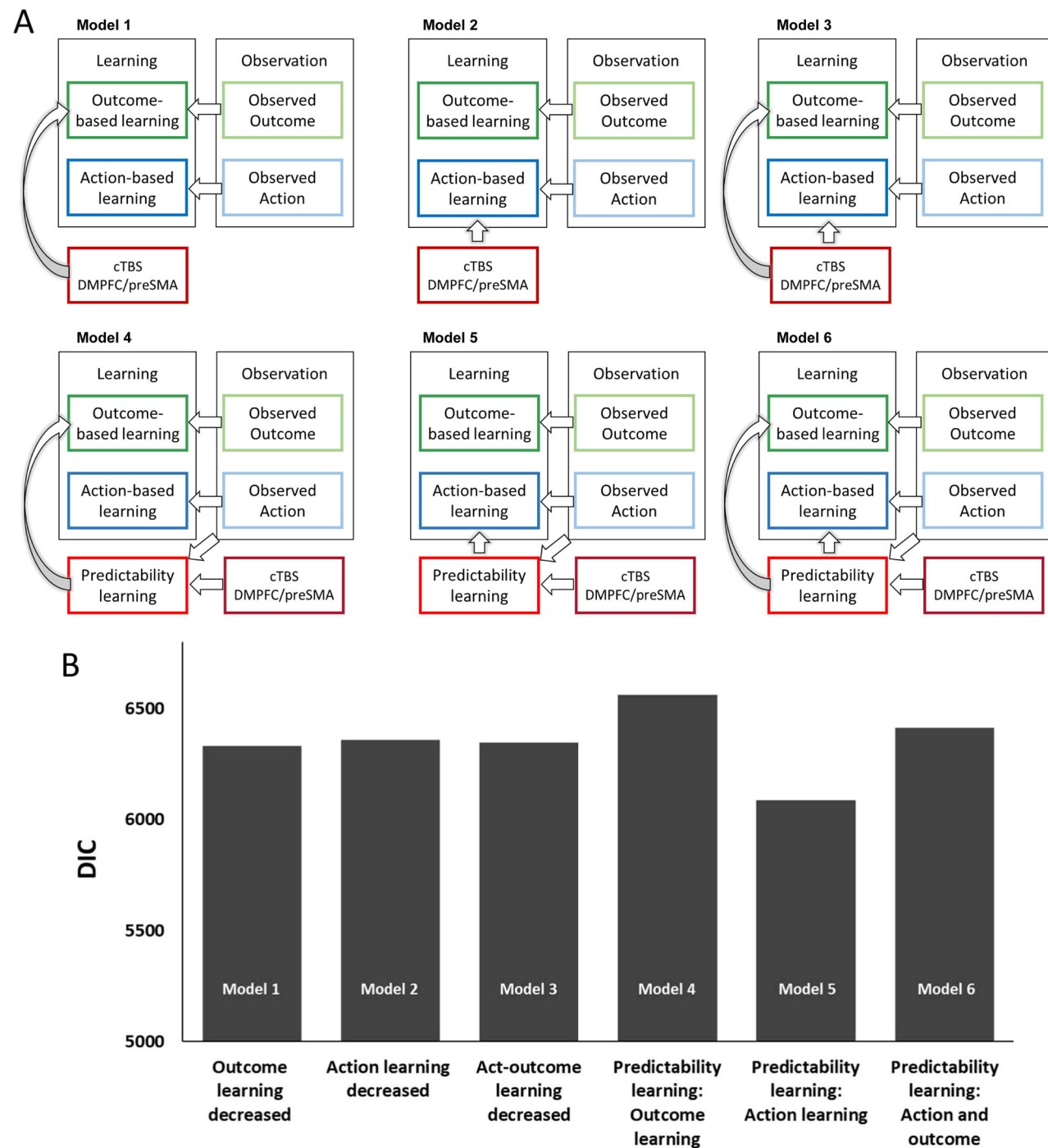

**Fig. 3 | Models and model comparison for choice for self in the decision phase. A** A schematic of the compared models. **B** Model comparison for choice for self. Lower DIC values indicate better model fit. The model with predictability learning controlling action-based learning (Model 5) explained choice for self best.

affected by DMPFC/preSMA cTBS. According to the models in this group, learning takes place in two stages, 1) learning about the predictability of demonstrator actions 2) learning from outcome and action prediction errors, with the degree of learning modulated by 1). Models 4-6 commonly assumed that DMPFC/preSMA cTBS affected predictability learning, but differed in whether the learned demonstrator predictability affected only observational outcome learning (Model 4, Predictability learning-Outcome model), only observational action learning (Model 5, Predictability learning-Action model) or both forms of learning (Model 6, Predictability learning-both model). In these models, the predictability of demonstrators could be assessed

through an internal simulation of demonstrator behavior. By simulating observed behavior over trials, the observers learned to what degree demonstrator choices became more predictable (by accumulating the observed choice predictability, see Methods), which modulated action- and/or outcome-based learning in the models of the second group. The first (Models 1-3) and the second (Models 4-6) group of models can have similar predictions at the group level, but the underlying cognitive processes are different. We fitted *choice for self* and *prediction of demonstrator actions* separately (to keep parameter space manageable) and attempted to find the best model for each behavior.

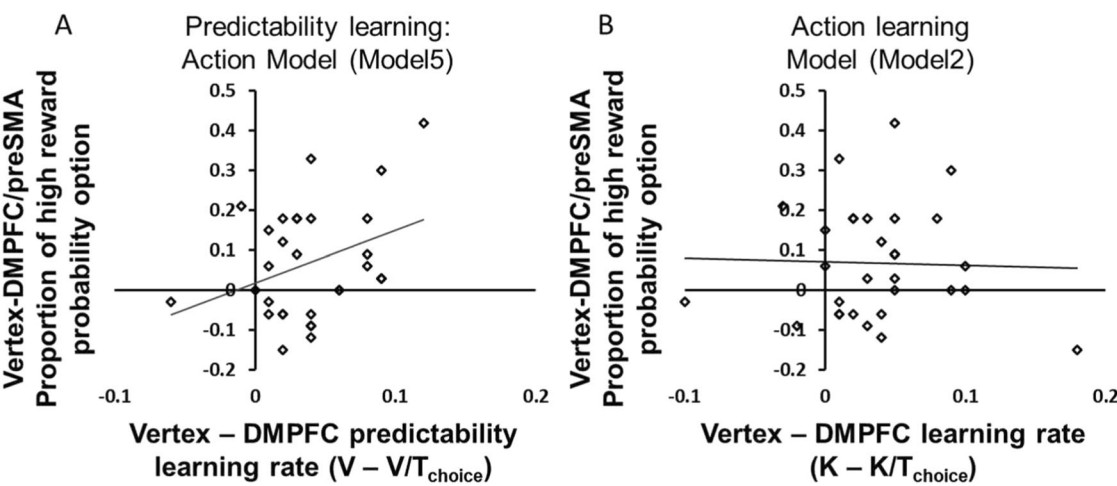

**Fig. 4 | Relation between stimulation effects in model-based and model-free analyses. A** The difference in predictability learning rates in vertex and DMPFC/preSMA cTBS conditions positively related to the difference in the proportion of choosing a high-reward probability option between vertex and DMPFC/preSMA

cTBS ($r = 0.340$, $p = 0.033$, one-tailed) for Model 5 (Predictability learning- Action). In contrast, (**B**) there was little relation for the equivalent analysis with Model 2 (Action learning, $r = -0.024$, $p = 0.452$, one-tailed).

### DMPFC/preSMA downregulation disrupts learning about demonstrator predictability for *choice for self*

Importantly, model comparison for *choice for self* data from all observational conditions indicated that the Predictability learning-Action model (Model 5) provided the best account for the behavioral pattern across vertex and DMPFC/preSMA cTBS (Fig. 3B). In this model, DMPFC/preSMA downregulation affected learning about demonstrator predictability primarily in the Action-Only condition. Simulations validated that this winning model can in principle uniquely identify the parameters that generated the observed behavioral pattern (Table S1). Together, these findings suggest that the winning model allows us to separate cTBS effects on different components of observation learning, and that the DMPFC/preSMA plays a central role in learning about demonstrator predictability from observed actions alone. Accordingly, the model comparison results not only corroborated but also expanded the findings of the model-independent analysis by pinpointing the mechanisms by which DMPFC/preSMA cTBS impaired action-based learning.

In addition to model comparison, we cross-checked whether the model captured the behavioral pattern we found in model-free analyses. First, we tested if the stimulation parameter $\tau_{choice}$ indeed captured the behavioral difference between the DMPFC/preSMA and vertex condition. A one-sample t-test showed that the stimulation parameter $\tau_{choice}$ was significantly larger than 1, $t(29) = 5.83$, $p < 0.001$, Cohen's $d = 0.68$, 95% CI = [0.47 0.98] indicating a significant DMPFC/preSMA cTBS effect. Second, we assessed whether the stimulation parameter was related (across participants) with the stimulation-induced behavioral change. This was confirmed by a significant correlation (Pearson's $r = 0.340$, $p = 0.033$, one-tailed as we expected only positive correlation; Fig. 4A) between the TMS-induced change in predictability learning rate and the performance difference between vertex and DMPFC/preSMA stimulation, in the Action-Only condition. Thus, the stimulation parameter $\tau_{choice}$ in the Predictability learning-Action model captured the stimulation-related variation in behavior as intended. By contrast, we did not find such a correlation ($r = -0.024$, $p = 0.454$, one-tailed as we expected only positive correlation, Fig. 4B) for the Action learning model (Model 2) that examined a decrease of action-based learning without incorporating predictability. This suggests that the Predictability learning-Action model provides an adequate account of the mechanisms underlying the behavioral effects induced by TMS.

Moreover, the stimulation parameter $\tau_{choice}$ also showed a negative correlation with the proportion of high reward probability option choice in the DMPFC/preSMA condition, Pearson's $r = -0.520$, $p = 0.020$, one-tailed as we expected only positive correlation (Figure S7A). In contrast, the predictability learning rate in the vertex condition was not correlated with performance (Figure S7C), Pearson's $r = -0.184$ $p = 0.330$. Thus, our results support the conclusion that the stimulation parameter in the model successfully captured the behavioral effects of stimulation.

### DMPFC/preSMA cTBS disrupts predictability learning for *prediction of demonstrator actions*

To understand whether cTBS has similar effects on the choices of participants for themselves and their predictions of demonstrator actions, we fitted the models that we used to explain *choice for self* also to *prediction of demonstrator actions* in all observational conditions. The model that best explained *prediction of demonstrator actions* was again the Predictability learning-Action model (Model 5), in which DMPFC/preSMA downregulation affected the processing of learned demonstrator predictability primarily in the Action-Only condition (Fig. S5). Thus, the same model best explained *choice for self* and *prediction of demonstrator actions*, in line with the notion that a common process underlies both types of behaviors.

Similar to *choice for self*, we ascertained for the Action-Only condition with the superb demonstrator that the stimulation parameter $\tau_{prediction}$ of the best model accounted for the cTBS effect on the proportion of correct *predictions of demonstrator actions*. The corresponding correlation was positive, Pearson's $r = 0.354$, $p = 0.026$, one-tailed (Fig. S7D). In the same condition, we also assessed the correlation between the difference of predictability learning rates between the vertex and the DMPFC/preSMA cTBS and the difference in the proportion of correct *predictions of demonstrator actions*. The correlation was not statistically significant, Pearson's $r = 0.09$, $p = 0.311$, one-tailed as we expected only positive correlation (Fig. S7C). This finding is compatible with the notion that the process disrupted by DMPFC/preSMA stimulation is expressed in both choices and predictions.

### Discussion

Using TMS, our study elucidates the causal role of DMPFC/preSMA in observational learning and pinpoints the computational process modulated by brain stimulation. Comparing DMPFC/preSMA cTBS effects between the different learning conditions, we find that DMPFC/

preSMA activity is crucial for observational learning only when observers have to rely on observed actions, but not when they can rely on observed outcomes. This role for observational learning seems important both for predicting the observable actions of others and for using the learned information to control the choices of the observers, as DMPFC/preSMA downregulation affected both functions. Computational modeling revealed that DMPFC/preSMA downregulation interfered with learning of demonstrator predictability over trials, especially when the predictability learning concerned actions of the demonstrator. Thus, the model-based analyses expanded upon the model-free analyses and highlighted a specific process that seems central for the involvement of DMPFC/preSMA in observational learning.

Our study shows a causal role for DMPFC/preSMA in observational learning and reveals that this role comprises specifically weighting the observational action-based learning rate with the predictability of others' actions. We were able to dissociate these two possibilities thanks to systematic assessment of two groups of models with a common structure (group 1: Models 1, 2, 3; group 2: Models 4, 5, 6) that predicted similar behavioral patterns at the group level (Model 1 ≈ Model 4, Model 2 ≈ Model 5, Model 3 ≈ Model 6), but based on cTBS effects on different underlying cognitive processes. The distinguishability of Models 2 and 5 was corroborated by the fact that the stimulation parameters from Model 5 but not those of the seemingly similar Model 2 correlated with individual differences between vertex and DMPFC/preSMA cTBS conditions. Thereby, we could show that DMPFC/preSMA downregulation not simply causes decreased action-based learning but decreased action predictability learning, which in turn modulates action-based learning. We establish this not just at the group level, but also by showing that performance correlates across participants with the individual TMS effects on specific model parameters.

Action predictability learning is a concept that can be linked to the role of the DMPFC in performance evaluation[14,20]. DMPFC has been functionally associated with monitoring and adjusting performance in response to behavioral errors of the agent themselves[14,22,23,24], observed errors of others[13] and irrelevant information evaluating one's ability[25]. In keeping with a previous suggestion[11], we propose that action predictability provides a formal measure of demonstrator performance. The predictability of the demonstrator should be particularly relevant and diagnostic for demonstrator quality when only actions are observable[26]. Our findings suggest that in such situations, the role of the DMPFC in using and optimizing observed information is relatively specific. The best fitting model indicates that learning about the predictability of demonstrator actions enables subsequent performance adjustment by dynamically setting the learning rate when weighting observed action prediction errors. Thus, our findings provide causal evidence for the role of DMPFC in performance monitoring and adjustment during social learning.

Our computational findings suggest that observational learning involves different sets of processes that may be related in a hierarchical fashion: Learning about demonstrator predictability and learning about the actions and outcomes of that demonstrator. This insight goes beyond previous computational models, which assumed a one-stage process with fixed learning rates regardless of demonstrator quality[8–10]. A recent study suggested that observational learning requires arbitration between emulation and imitation[11], but no model corroborated a hierarchical structure where learning about the properties of a demonstrator governs learning about observed actions with a link to a neural substrate. Given that learning from others varies with demonstrators as well as observed information[8,26–28], it seems reasonable to assume the existence of a neural substrate that modulates learning about evaluated predictability. However, direct evidence for this neural process has been sparse. Our study now provides evidence that DMPFC/

preSMA is causally involved in learning (or evaluating) the predictability of observed actions emitted by a superb demonstrator.

Another interesting finding is that in the best predictability learning model, predictability modulated the impact of observed actions but not the impact of observed outcomes. The previous literature has been inconclusive about whether the DMPFC/preSMA is more closely related to action or outcome learning and reported responses to both (negative) outcomes and (erroneous) actions[22] as well as the associated prediction errors[10]. By directly comparing cTBS effects on both these proposed mechanisms, our results provide evidence for a more central role of DMPFC/preSMA for action monitoring.

Downregulation of DMPFC/preSMA decreased predictability learning and its effect appeared in both prediction and choice. The separation of tracking the behavior of the demonstrators and using that information for choice allowed us to infer that the monitoring process in the DMPFC/preSMA may not necessarily be linked to a specific type of behavioral adjustment, thereby qualifying previous suggestions[29]. The *prediction of demonstrator actions* in our task corresponds to unrewarded tracking of information while the *choice for self* aims to maximize reward for oneself. The common effects of DMPFC/preSMA downregulation suggest that the monitoring and performance adjustment processes of DMPFC contribute to both functions, regardless of decision type.

Some studies have demonstrated that the MPFC contributes to the distinction between self and other[30,31], and that downregulation of the DMPFC leads to self-other mergence when evaluating the abilities of oneself and other people[32]. Our study focused on learning from observing the actions of others, but also formally tested if DMPFC stimulation changes learning from outcomes for oneself. We find that learning from own outcomes (individual learning condition) is not affected by DMPFC/preSMA cTBS, suggesting that the stimulation effects are specific to learning from observing (the actions of) others. Thus, possible merging of (the outcome representations of) self and other as a consequence of stimulation is unlikely to have caused the effects on observational learning in the present study. However, our study targeted a more posterior part of DMPFC compared to the studies of self-other mergence[32,33]. Thus, it is well possible that different portions of the DMPFC govern both functions. Moreover, the precise area in dorsal and ventral MPFC distinguishing self from other may depend on individual traits[34,35].

It is noteworthy that there is ambiguity when it comes to the precise anatomical labelling of the site we stimulated, as the term "DMPFC" refers to a large brain region in superior frontal gyrus. Our stimulation site was based on coordinates from a previous study[10]; it is located at the border between Broadman areas 6 and 9 in a posterior part of mPFC. The terminology used in other studies for this area ranges from preSMA[11] to DMPFC[8,10,36,37], posterior mPFC, and dorsal anterior cingulate cortex/rostral cingulate zone[38–40]. Due to the complexity of a precise definition, we opted to refer to the stimulation site as "DMPFC/preSMA", indicating that assignment of a single and precise anatomical label is difficult.

The model recovery confirmed that the models were identifiable for *choice for self* but there was a bit weaker distinction for models for prediction of demonstrator actions. This may be due to the difference between the range of fitted parameters for the models for *choice for self* and *prediction of demonstrator actions*[41]. However, as we applied the same models for both behaviors, the model recovery results for *choice for self* provide evidence that these models are clearly identifiable in principle. Importantly, the model selection was entirely based on *choice for self* and the corresponding model recovery is the only one that matters for the conclusions of the study.

As we used offline rather than online TMS, we could not answer whether the DMPFC/preSMA implements only performance monitoring or also performance adjustment. DMPFC activity had been related

to behavioral switching and control based on recently monitored information[14,22,42,43], in line with an additional role in adjustment. However, rather than controlling behavior directly, DMPFC could simply monitor performance and send the arising error information to other brain regions that are critical for controlling behavior such as LPFC[23,44–46]. Future research may want to dissociate these two possibilities by stimulating online, either before prediction or before choice, and/or by combining cTBS with neuroimaging measures.

In conclusion, our computational brain stimulation study showed that the DMPFC/preSMA is essential for observational learning by monitoring the predictability of the demonstrator and adjusting the weights given to observed action prediction errors. This finding provides a more unified framework to comprehensively explain the multifaceted function of the DMPFC/preSMA in observational learning, performance evaluation and performance modulation based on observational learning. We show that learning from observed actions arises from a multi-stage process that learns about demonstrator quality and weighs action prediction errors accordingly.

## Methods

### Participants

Thirty-one healthy volunteers (18 females, $M_{age} = 23.59 \pm 2.45$) were recruited from the University of Zurich and the Swiss Federal Institute of Technology in Zurich. They came to the designated TMS room of the Laboratory for Social and Neural Systems Research (SNS Lab) of the UZH Department of Economics, located at the University Hospital Zurich (USZ), for two sessions (active and control TMS). As one participant did not complete the experiment, the data were not included in the analyses. Power analyses indicated that a minimum of thirty participants was needed to obtain a statistical power of 80% for detecting significant TMS effects (assuming a standard significance criterion of $\alpha = 5\%$) in a within-subject design, with the same moderate effect size ($d = 0.67$) of previous TMS research[47]. Eligibility for TMS was determined based on standardized safety guidelines[48]. Participants received a fixed CHF 40/hour and a performance-dependent (CHF $17 \pm 0.80$) payment. All participants gave informed written consent prior to participation. The study protocol was approved by the Research Ethics Committee of the canton of Zurich. The investigation was conducted in full accordance with the principles expressed in the Declaration of Helsinki.

### Task

We used an observational learning task similar to those of our previous studies[8,9]. The task consisted of four observational learning conditions (two types of information: Action-Outcome and Action-Only, crossed with two demonstrator types: superb and bad) and an individual learning condition. In each of these conditions, the participants had to learn to identify and select the more frequently-rewarded of two fractal images. The two fractals were associated with different reward probabilities (70% vs. 30%), which remained constant throughout each condition. Before choosing one of the two images, participants observed a demonstrator choosing between the same two images. They observed only one demonstrator in each block. Participants were told that the demonstrators were participants of a previous study who had performed the same two-armed bandit task for themselves (i.e. individual learning) with the same pair of images. Accordingly, participants could use this information to inform their own choice of the fractal image associated with the higher reward probability.

Each trial comprised an observation phase (i.e. observing the demonstrator making a decision) and a decision phase (i.e. participants choosing for themselves; Fig. 1A). The first trial of each block did not include an observation phase to prevent a potential bias for one of the two fractal images. To avoid confusion between the two phases, the screen was vertically divided into two halves, with one half dedicated to the demonstrator and the other half to the participant. The

assignment of agents to screen halves remained constant for each participant but was counterbalanced across participants.

At the beginning of the observation phase, a photo of the face of the demonstrator was displayed for 1 s. After that, two fractal images appeared underneath the photo of the demonstrator and participants had 2 s to predict which fractal the demonstrator would choose; this prediction was indicated with a white frame (0.5 s). Then, the option that the demonstrator chose was indicated with a green frame for 1 s. Depending on the condition, participants next also observed the actual outcome of the demonstrator (Action-Outcome condition) or a pixel-matched scrambled image of that outcome (Action-Only condition) for 1.5 s.

The decision phase began with the word "you", shown for 1 s on the participant side of the monitor, followed by the same two fractals the demonstrator had decided between in the observation phase. Within 2 s, participants chose one of the two fractals for themselves. The chosen fractal was framed for 0.5 s. Finally, participants saw a scrambled outcome image (0.5 s) which prevented individual learning during the observational task. The location of the two fractal images was determined randomly at the beginning of each phase to ensure that behavior was based on the identity of the fractal images rather than their location. Learning was incentivized by paying out the total points the participants earned (conversion rate 100 points = 1CHF) at the end of the experiment.

We used the two conditions (Action-Outcome vs. Action-Only) to disentangle action-related observational learning from outcome-related observational learning. In the Action-Outcome condition, participants could learn from observed outcomes and/or from observed actions. By contrast, in the Action-Only condition any learning had to be based on observed actions. However, in the action-outcome condition, the two conditions could fail to disentangle observational outcome learning from observational action-based learning or simple imitation because, at least in principle, participants could rely exclusively on observed actions also in the Action-Outcome condition. Although this possibility seemed unlikely, we simulated both a superb demonstrator who learned well (70-90% choices of higher reward probability option per block, learning rate ($\alpha$) of 0.3 as modeled by a standard Q-learning model) and a bad demonstrator who learned nothing (50% choices of higher reward probability option per block). In the Action-Outcome condition with the bad demonstrator, only outcome-based learning but not imitation or action-based learning allows observers to perform better than chance. Moreover, in the Action-Only condition participants may realize that the behavior of the non-learning demonstrator is random and therefore gradually reduce imitation whereas pure imitation would not be associated with such a reduction.

In addition to the observational learning conditions, participants completed an individual learning condition (Figure S2) that also consisted of an observation phase and a decision phase for every trial. However, the observation phase provided no relevant information: The photo of the demonstrator was replaced by a scrambled image, both options were framed at the usual time of demonstrator choice, and a scrambled outcome was shown. Instead, participants now saw the outcomes of their own choices during the decision phase. The task comprised three sessions with two blocks of eleven trials per condition, resulting in a total of 330 trials. The order of the conditions within a session was randomized.

**Face stimuli.** Face stimuli representing demonstrators were chosen from a face database[49]. We selected young (i.e. 20–30 years old) Caucasian faces with a neutral expression. To prevent performance and impression carryover effects induced by the demonstrator, we used one facial image per block, resulting in twenty-four images (three blocks x four social learning conditions x two visits). These images were grouped into two sets with similar attractiveness as judged by a group of independent raters[49] and each set was randomly assigned to

one of the visits, counterbalanced across participants. The sex of the demonstrators was matched to that of the participants.

**Fractal images.** We generated sixty fractal images (ChaosPro fractal generator, http://chaospro.de) and separated them into two sets of fifteen pairs of images (three blocks x five learning conditions). Each set was randomly assigned to one of the visits and counterbalanced across participants. A different pair of images was used for each block. The images of each pair were matched on pleasantness based on the rating of an independent sample ($n = 10$) to minimize bias in early trials. Participants in the TMS study also rated the fractal images used in the observational learning task prior to the TMS experiment; there was no significant difference between ratings for any pair of fractal images, t(29) = 1.149, $p = 0.29$.

**Questionnaires.** Participants filled in questionnaires at the beginning of their second visit. These questionnaires measured traits related to social behavior such as the Interpersonal Reactivity Index[50], and Machiavellianism[51] as well as non-verbal intelligence, operationalized with the Raven test[52]. To explore whether individual differences in social or cognitive traits could explain individual differences in observational learning, we performed correlation analyses. In particular, we related the Interpersonal Reactivity Index, Machiavellianism, and non-verbal intelligence to observational learning in all conditions and to observational learning differences between vertex and DMPFC stimulation. We found no significant relations, all ps >0.068 (Table S2).

**TMS**
Participants were stimulated over the DMPFC/preSMA and, in a separate session, over the vertex with standard continuous theta-burst stimulation (cTBS[53]). We used a Magstim Rapid2 stimulator (Magstim Co.) with a figure-of-eight coil (80 mm diameter of each winding) for brain stimulation. The stimulation sites were localized on individual T1-weighted structural scans using Brainsight frameless stereotaxy (Rogue Research). To determine the stimulation intensity, we first measured resting and active motor thresholds using electromyographical assessments implemented in the Brainsight system. The resting motor threshold (rMT) was defined as the lowest single-pulse intensity that elicited motor evoked potentials (MEP; >200 mV in amplitude) in at least five of ten pulses from the contralateral first dorsal interosseous muscle while resting the hand on a pillow. The active motor threshold (aMT) corresponded to the lowest single-pulse intensity evoking an MEP higher than 200 mV in five of ten trials while the participant's thumb and index finger were touching with constant pressure of about 20% maximum force. The stimulation intensity was set to 80% of aMT with distance correction[54] to compensate for differences in distance from skull to motor cortex and skull to the stimulation site (mean stimulation intensity: 40.55 ± 5.69% s.e.m. of maximum intensity).

The target coordinate for DMPFC/preSMA stimulation was taken from a previous fMRI study that reported increased DMPFC/preSMA activity in an observational learning task (x = 6, y = 14, z = 52[10]). We transformed this coordinate into the native space of each participant's structural scan using the inverse parameter estimates for spatial normalization of the anatomical scan performed in SPM12. The vertex served as a control site and was defined for each participant as the junction of the pre- and post-central sulcus in the intrahemispheric fissure.

Standard cTBS[55] was applied for 40 s to DMPFC/preSMA or vertex. Specifically, bursts of 3 pulses at 50 Hz were repeated with a frequency of 5 Hz, resulting in a total of 600 pulses. The figure-of-eight-coil was positioned parallel to the midline with the handle pointing backward and tangential to the brain surface. It has been shown that 40 s of cTBS can reduce the excitability of the stimulated brain region for about 60 min[53]. Given that the average duration of the observational learning task was 21.43 min (M = 21.43 ± 0.41; min. 20.71- max. 22.58), we could

be reasonably confident that the TMS protocol reduced excitability of the stimulated region during the full period of task performance. Participants received one or two test pulses over the session-specific stimulation site before actual cTBS stimulation to examine whether stimulation caused muscle twitching or discomfort. The experiment continued only if participants were comfortable with the stimulation. The participants received cTBS stimulation while sitting in front of the task computer and started the task immediately after the stimulation. The same procedure was repeated for the first and the second visit and the order of brain stimulation sites was counterbalanced across participants (i.e. fifteen participants received vertex stimulation first and the other fifteen received DMPFC/preSMA stimulation first). Visits were one to three weeks apart.

**Behavioral data analyses**
We examined the effects of cTBS on the proportion of choosing the higher reward probability option, the proportion of *correct predictions of demonstrator actions*, *choice for self* response times, *prediction of demonstrator actions* response times and imitation behaviors. We examined the fixed effects of stimulation site (vertex, DMPFC/preSMA), demonstrator quality (superb, bad), observed information (Action-Outcome, Action-Only) and their interactions in separate generalized linear mixed models for all dependent variables. The models included participant-specific random intercepts and random slopes for the factors stimulation site, demonstrator quality and observed information. In additional models, we included stimulation order as a control regressor (all the results are qualitatively the same and significant or non-significant when controlling for stimulation order). To assess the effects of cTBS in a model-free but sensitive manner, we performed planned pairwise comparisons for the effect of stimulation site (vertex, DMPFC/preSMA).

All statistical analyses were performed in R (v3.7.1; www.r-project.org). All linear mixed models were analyzed with the "lme4" and "optimx" package. We also computed Bayes factors (BF) with the "BayesFactor" package to test how strongly the data favored the alternative compared to the null hypothesis ($BF_{10}$) based on BICs[56].

**Computational model for *choice for self***
To specify the effects of DMPFC/preSMA downregulation on observational learning from the actions of the superb demonstrator, we employed a computational approach. Two groups of models tested 1) DMPFC/preSMA cTBS effects on outcome- and/or action-based learning and 2) DMPFC/preSMA cTBS effects on learning of demonstrator predictability, which in turn affected outcome- and/or action-based learning. Our models were based on a reinforcement learning framework that aimed to reduce outcome prediction errors and/or action prediction errors. First, we describe the aspects shared by all the models and then explain the specific variations from the basic model. Note that the Figure S6 provides an overview of the relations of Equations in each model and the parameters in the models were introduced in Tables S3 and S4.

Models of observational learning make inferences from the perspective of the demonstrator[10], a tradition we followed in our approach (Eqs. (1) and (2)). The simulation followed standard reinforcement learning where Q values for different actions are updated by a prediction error modulated by a learning rate. To convert Q values to binary choice, we used a standard softmax function (Eq. (2)). We performed a grid search to systematically test different combinations of demonstrator learning rates ($\alpha_{demonstrator}$, range 0.1–1) and choice temperature ($\beta_{demonstrator}$, range 1–10). Based on the best fitting parameter values (Tables S5 and S6), we fixed $\alpha_{demonstrator}$ to 0.2 and $\beta_{demonstrator}$ to 9 for subsequent model estimations.

$$Qa_{(i)}^{demonstrator} = Qa_{(i)}^{demonstrator} + \alpha^{demonstrator} * \delta_{(i)}^{demonstrator} \qquad (1)$$

$$p^{demonstrator}(a) = \frac{1}{1 + \exp\left(-\beta^{demonstrator}\left(Qa_i^{demonstrator} - Qb_i^{demonstrator}\right)\right)} \tag{2}$$

Observational learning was based on simulated outcome and action prediction errors during the observation of the demonstrator's action and outcome. Learning from observed outcomes was implemented as value update via simulated outcome prediction errors (Eq. (3), Action-Outcome), which corresponded to the difference between the outcome the demonstrator actually received and the value of the chosen option the observer should have simulated. Learning from observed actions was updated with action prediction errors, implemented as the difference between the demonstrator's actual choice and the choice the participant should have expected (Eq. (4); Action-Outcome and Action-Only). In the Action-Only condition, accumulated probability AP(a)$_i$ captured the expected probability to choose option a. In the Action-Outcome condition, learning from observed outcomes and observed actions was combined to calculate the simulated value of option a, via a weighting parameter that arbitrated between the predictions arising from the two pieces of information (Eq. (5)). The probability that the participant's choices were guided by this simulated value was calculated with a softmax function that included a perseverance parameter ρ (Eq. (6))[57]. C is the choice of previous trial. The following equations give the basic structure of the observational learning models we tested; all models were variations of the basic model but tested different hypotheses concerning the effects of cTBS.

$$Q(a)_i^{outcome} = Q(a)_i^{outcome} + \alpha * \left(R_i^{demonstrator} - Q(a)_i^{outcome}\right) \tag{3}$$

$$Q(a)_i^{action} = Q(a)_i^{action} + k * \left(A_i^{demonstrator} - Q_i^{action}\right) \tag{4}$$

$$V(a)_i = w * Q(a)_i^{outcome} + (1 - w) * Q(a)_i^{action}$$
$$w = 0, \text{in Action} - \text{Only condition} \tag{5}$$

$$p(a) = \frac{1}{1 + \exp(-\beta(Va_i - Vb_i) + \rho * (Ca - Cb))} \tag{6}$$

We built two groups of models: Models 1–3 tested whether DMPFC/preSMA cTBS directly decreased outcome- and/or action-based learning; the other group (Models 4–6) examined whether learning of the demonstrator predictability modulated outcome- and/or action-based learning.

### DMPFC/preSMA cTBS decreases learning from observed outcomes and/or actions (Models 1-3)

The Outcome model (Model 1) included the vertex condition as a baseline (Eq. (3)) and captured the effect of stimulation with a parameter that quantified the extent to which DMPFC/preSMA cTBS decreased learning from observed outcomes. Specifically, the stimulation parameter τ in Eq. (7) modulated the outcome-based learning parameter κ in the DMPFC/preSMA condition.

$$Q(a)_i^{outcome} = Q(a)_i^{outcome} + \frac{\alpha}{\tau}\left(R_i^{demonstrator} - Q(a)_i^{outcome}\right) \tag{7}$$

τ ranged between 0 to 10, allowing for both stimulation-induced impairments and improvements of observational learning. The other parameters ranged between 0 and 1, with the exception of beta and rho, which ranged [0 30] and [0 3] respectively. We used identical parameter ranges in all the models.

The Action learning model (Model 2) tested whether DMPFC/preSMA cTBS decreased learning from observed actions. It varied Eq. (4) of the basic model by adding a stimulation parameter τ that modulated action-based learning parameter κ in the DMPFC/preSMA condition (Eq. (8))

$$Q(a)_i^{action} = Q(a)_i^{action} + \frac{k}{\tau} * \left(A_i^{demonstrator} - Q(a)_i^{action}\right) \tag{8}$$

In the Action-Outcome model (Model 3), outcome- and action-based learning could both be decreased by DMPFC/preSMA cTBS. In turn, this model included Eq. (7) and (8) which incorporated τ parameters reflecting the cTBS effect, instead of Eq. (3) and (4).

### DMPFC/preSMA cTBS decreases learning about demonstrator predictability (Models 4-6)

Models 4-6 extended the basic model by allowing that learning about demonstrator predictability can modulate outcome- and/or action-based learning and assessed whether DMPFC/preSMA cTBS disrupted learning about demonstrator predictability. A substantial literature has shown that the DMPFC is involved in error detection, reward volatility, response conflict, and the updating of action value[14,20]. A common denominator of these functions is performance monitoring. When monitoring or inferring performance quality of the demonstrator, it is crucial to keep track of the predictability of the monitored actions. In line with this notion, the findings from the model-free analysis suggested that DMPFC/preSMA plays a central role particularly in monitoring observed performance when predicting demonstrator choice in the Action-Only condition of our observational learning task. We therefore reasoned that predictability should be driven by information that participants could observe from the demonstrators' actions. To capture predictability formally, we first computed Shannon's entropy (Eq. (9)) as the degree of simulated uncertainty entailed by demonstrator actions in the n$_{th}$ trial[58]. Then, we subtracted entropy from 1 to obtain predictability (Eq. (10)). The predictability of the demonstrator in the n$_{th}$ trial was updated by the difference between the expected (accumulated) predictability and the actual predictability on that trial. Accumulated predictability (AcP) reflected the predictability of a demonstrator across trials in a session and was used to modulate action-based learning (Eq. (11)). Potential effects of DMPFC/preSMA downregulation on predictability learning were captured by τ$_{choice}$ in Eq. (12).

$$H(x) = -\sum_{i=1}^{n} P^{demonstrator}(x_i) \log P^{demonstrator}(x_i) \tag{9}$$

$$predictability = 1 - H(x) \tag{10}$$

$$AcP_{(i)} = AcP_{(i-1)} + \upsilon * \left(predictability_i - AcP_{(i-1)}\right) \tag{11}$$

$$AcP_{(i)} = AcP_{(i-1)} + \frac{\upsilon}{\tau_{choice}} * \left(predictability_i - AcP_{(i-1)}\right) \tag{12}$$

Models 4-6 all had the predictability update process in common, but differed in how learning from observed outcomes and actions was modulated by learned demonstrator predictability. Specifically, in the Predictability learning-Outcome model (Model 4), observational outcome learning could be modulated by learned predictability (Eq. (13) instead of Eq. (3)). This model allowed for outcome-based learning to be affected by DMPFC/preSMA cTBS-induced reductions in

predictability learning.

$$Q(a)_i^{outcome} = Q(a)_i + AcP_i * \left( R_i^{demonstrator} - Q(a)_i^{action} \right) \quad (13)$$

In the Predictability learning-Action model (Model 5), observational action learning was modulated by learned predictability (Eq. (14) instead of Eq. (4)). According to this model, action-based learning could be decreased by DMPFC/preSMA cTBS through the decrease of predictability learning.

$$Q(a)_i^{action} = Q(a)_i^{action} + AcP_i * \left( A_i^{demonstrator} - Q(a)_i^{action} \right) \quad (14)$$

In the Predictability learning-both model (Model 6), both forms of observational learning could be modulated by learned predictability (Eqs. (13) and (14) instead of Eqs. (3) and (4)). By extension, both outcome- and action-learning could be affected by decreased predictability learning in the DMPFC/preSMA cTBS condition.

As a sanity check, we compared the predictability of the superb and bad demonstrator (starting from the 4th trial, as predictability in earlier trials is similarly low between the conditions due to the low number of observations). Predictability of the superb demonstrator was significantly higher than the one of the bad demonstrator, $t(2519) = -2.80$, $p = 0.005$. Also, trial-by-trial predictability as used in the computational model (see Methods) was negatively correlated with trial reaction time, $r = -0.101$, $p < 0.005$, indicating that when predictability was higher, participants responded more quickly.

To assess the relation between the learning rate $\upsilon$ and $\tau_{choice}$, we performed robust regression analyses and found no significant correlation between $\upsilon$ and either $\tau_{choice}$ (coefficient = 0.24, $p = 0.20$) or $\tau_{prediction}$ (coefficient = 0.235, $p = 0.224$). In addition, the multiplied values of the two parameters were within the normal range for learning processes (0-1), both for choice and prediction, indicating that there was no excess in the predictability learning process, or, by extension, in the resulting value estimate.

## Computational model for prediction of demonstrator action

As the model-independent analysis suggested that DMPFC/preSMA stimulation decreased prediction accuracy also primarily in the Action-Only condition with the superb demonstrator, we examined whether the effects of TMS on *prediction of demonstrator actions* and *choice for self* reflect one single mechanism. Even though the essential components of observational learning (i.e. tracking demonstrator choice) are necessarily shared in *choice for self* and *prediction of demonstrator actions*, it is worth keeping in mind that the goal of *prediction of demonstrator actions* (i.e. to accurately foretell demonstrator behavior) differs from the goal of *choice for self* (i.e. to maximize observer earnings). Thus, it is also possible that different mechanisms underlie these two forms of behavior. However, we fitted the participant *prediction of demonstrator actions* with Models 1 to 6 (assessing one common mechanism) to show that the same components influenced by TMS during *choice for self* was also affected during the *prediction of demonstrator actions* and found largely similar effects, as described in the main text.

## Parameter estimation, model comparison and validity checks

We used a hierarchical Bayesian approach for model estimation. The hierarchical procedure assumes that the model parameters of individual participants are drawn from a group-level distribution[59]. This allows more reliable individual-level model estimation than separate estimation for each participant, as the individual-level parameter estimates are constrained by the group distribution. Group-level distributions for all model parameters were assumed to be beta distributions. The group-level means were assigned a uniform hyperprior on the interval [0.001,0.999] and the group-level precisions were assigned a uniform hyperprior in the range [0.001, 10]. Beta

distributions are typically defined by two shape parameters (a, b). We parameterized a = group level mean*group-level precision, b = (1- group level mean)*group-level precision[60] and transformed the range of beta, rho, tau parameters to [0-30], [0-3], [0-10] at the individual-level. We inferred posterior distributions for all model parameters using Markov chain Monte Carlo (MCMC) sampling, as implemented in the JAGS package via the R2jags interface[61]. We ran three independent MCMC chains with different starting values per model parameter and collected 40,000 posterior samples per chain. We discarded the first 10,000 iterations of each chain as burn-in. In addition, we only used every 5th iteration to remove autocorrelation. Consequently, we obtained 18,000 representative samples per parameter per model. We used the Deviance Information Criterion (DIC) for the model comparisons[62]. The DIC is an index of the goodness of model fit, penalized by its effective number of parameters. A smaller DIC indicates a better model fit. The posterior distributions of the parameters are shown in Figure S10.

We ran parameter recovery analyses to assess whether our winning model (Predictability learning-Action model) was able to reliably estimate parameters used to generate simulated data. We simulated data from 30 participants at 30 different instances covering the parameter space and fitted the model to the simulated data. The correspondence between the simulated and recovered parameters was reasonably high, all rs >0.6 for *choice for self* (Table S1).

We also performed model recovery for each model to verify whether the models are distinguishable given the data and the fitting procedure. We ran 50 simulations of each model with the data of each participant and each trial. For the simulation, model parameters were chosen from the distribution of fitted parameters. After simulation, we fitted models on the data generated by each model and report the confusion matrix (Fig. S9). The model recovery confirmed that the models were identifiable for *choice for self* but there was a bit weaker distinction for models for prediction of demonstrator actions. This may be due to the narrow range of fitted parameters as shown in Fig. S10B. However, as we applied the same models for both behaviors, the model recovery results for *choice for self* provide evidence that these models are clearly identifiable in principle. Importantly, the model selection was entirely based on *choice for self* and the corresponding model recovery is the only one that matters for the conclusions of the study.

We performed a simulation of the winning Predictability learning-Action model. To do so, we simulated 30 different data sets 100 times using the estimated parameter distribution from fitting the data. Then, we compared data generated with a model with $\tau_{choice} = 1$ (no TMS effect) to those generated with a model with $\tau_{choice} = 5$ (disruptive effect of DMPFC stimulation; Fig. S8). This simulation showed that the difference in the $\tau$ parameter indeed was sufficient to generate an analogous effect of DMPFC/preSMA downregulation on *choice for self* as observed in the data.

Additionally, learning rates and softmax temperature often correlate (Krefeld-Schwalb et al. 2022) which can limit interpretability of the findings of learning models. We found little evidence for this concern in our data. Specifically, correlation analyses between $\beta$ and $\alpha$ as well as between $\beta$ and $\upsilon$ (learning rate in the best model) revealed no significant correlations (*choice for self*: $\beta$ and $\alpha$, $r = 0.252$, $p = 0.180$; $\beta$ and $\upsilon$, $r = -0.167$, $p = 0.377$; *prediction of demonstrator's action*: $\beta$ and $\alpha$, $r = -0.033$, $p = 0.861$; $\beta$ and $\upsilon$, $r = -0.253$, $p = 0.178$).

## Reporting summary

Further information on research design is available in the Nature Portfolio Reporting Summary linked to this article.

## Data availability

The raw and preprocessed data in this study is available at https://doi.org/10.17605/OSF.IO/FRUA8.

## Code availability

The code for the analyses is available at https://doi.org/10.17605/OSF.IO/FRUA8.

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

## Acknowledgements
We thank Anna Giaratana, Giorgia Bergmann, Hui-Kuan Chung, Hugo Fluhr, Jae-Chang Kim, Lydia Hellrung, Maike Brandt, Michele Garagnani, Nick Doren, Stephan Nebe, Susanna Gobbi, Tanja Müller, Viktor Timo-khov as well as the speakers of the Neuroeconomics seminar for valu-able discussions. We thank Cornelia Schnyder and the recruitment team for the help with recruitment of participants.

This study was funded by grants of the Swiss National Science Foundation (100014_165884 100019_176016, IZKSZ3_162109, and 10001C_188878) to P.N.T. and by the University of Zurich (K-33151-02-01) to P.K. A.S. received an Emmy Noether fellowship (SO 1636/2-1) from the German Research Foundation and a research grant from the Bohringer Ingelheim Foundation. B.L. was supported by a Wallenberg Academy Fellow grant from the Knut and Alice Wallenberg Foundation (KAW 2021.0148) and a Starting Grant (SOLAR ERC-2021-STG – 101042529) from the European Research Council. CCR received funding from the European Research Council (ERC) under the European Union's Horizon 2020 research and innovation programme (grant agreement No 725355, ERC consolidator grant BRAINCODES) and from the University Research Priority Program 'Adaptive Brain Circuits in Development and Learning' (URPP AdaBD) at the University of Zurich.

## Author contributions
P.K. and P.N.T. designed the experiments. P.K. and M.M. collected the data. P.K. analyzed the data with inputs from P.N.T., B.L., A.S., and C.C.R. P.K., C.C.R. and P.N.T. wrote the paper with inputs from B.L., A.S., and M.M.

## Competing interests
The authors declare no competing interests.
