## [Peer Review File · Nature Communications]

Reviewers' comments:

Reviewer #1 (Remarks to the Author):

Kang and colleagues report an interesting investigation of social learning using brain stimulation. They utilise a familiar paradigm that has been used by the authors and others in the past. They combine it with an offline brain stimulation protocol that disrupts dorsomedial prefrontal cortex (dmpfc) activity for a short period of time. The authors find stimulation effects on two outcome measures. After stimulation, participants predict other's choices less well and they are less well able to pick a higher-value option that they have to infer from others' choices. Both findings indicate that dmpfc stimulation disrupts learning from others and the impairments only occur when there is actually a relationship between the demonstrator's choices and the outcomes (i.e., only for a superb, not a random demonstrator).

In my view, the study is well done. Even though the effects of interest are not significant in omnibus tests, they are significant when directly comparing between the stimulation condition and a vertex control condition, which is critical. The tests are well justified by a priori hypotheses. Personally, I don't think the RL modelling and in particular the idea that dmpfc stimulation specifically affects "predictability" adds very much. That effect seems to occur because learning is only affected for the superb but not the random demonstrator – so only in situation where learning from the demonstrator makes sense. Nevertheless, methods wise, the RL modelling is very well done and thoroughly conducted using advanced fitting techniques, which is appropriate given the small number of trials per participants (due to the brain stimulation time restrictions). Also, the use of a non-social control condition is a strength of the paper.

Major points:

1. I strongly urge the authors to show a figure with the brain stimulation coordinates overlaid on a structural image of the brain in main paper. This is because the authors stimulate a region of the brain that is usually not referred to as "dorsomedial prefrontal cortex". It is a very posterior region close pre-SMA. The authors note that they refer to this region as dmpfc, because the paper they take the coordinates from talks of dmpfc. This is legitimate, but it is also necessary to acknowledge that rarely anyone else refers to this region as dmpfc. That is why I think there should be a figure showing clearly where in the brain the brain stimulation took place.
2. Model comparisons: The models are sufficiently described, but I had a very hard time figuring out how the model comparisons were conducted and in particular which data the models were fitted on. It was unclear which of the two decisions the data was fitted on (or both at the same time?). Also it was unclear if the whole sample was fitted together or not. The statement on page 12 "model comparison indicated that the Predictability-Action model (model 5) provided the best account for the behavioural effect of DMPFC cTBS" sounds like the model was specifically fitted on the differences between conditions?
3. Other recent accounts of learning about other people suggest that self-other-distinction is an issue (Ereira et al., 2018, 2020) and implicate specifically dmPFC in this ability, also using brain stimulation

evidence (Wittmann et al., 2016, 2021). Could the authors findings alternatively also be explained by failures of self-other-distinction induced by brain stimulation? This could be tested in the model comparison or at the very least discussed as another possibility.

Minor points

1. I strongly suggest to give a better intuition how the models work in the main text. They appear very obscure at the moment.
2. When I read the manuscript, I kept getting confused about which of the two decisions that are in a trial the authors talked about – the demonstrator-prediction or the individual-stimulus-choice. I suggest signposting this better throughout the manuscript and not just talking about “the choice” in parts of the results section.
3. A table displaying the free parameters per model could be useful to get an overview over the models.
4. Page 10/line 10: “The results remained qualitatively the same when controlling for stimulation order.” – Yes, but were the effects of interest still significant?
5. The section about “imitation” on page 10. I think the point here is that brain stimulation effects cannot be explained as changing people’s tendency to blindly imitate choices. If that is true, could the authors simply state that? I found the long explanation of what choice history is a bit misplaced and felt it did not help the authors make their point. Of course the authors may also keep it as is and try to make clearer what they mean.

Reviewer #2 (Remarks to the Author):

In this study, Kang and colleagues are investigating the impact of a repeated TMS stimulation of a structure of the dorsomedial prefrontal cortex in observational learning.

The authors chose a stimulation site based on a previous study by Suzuki and colleagues. The chosen site is also close to the cluster they identified in their previous study (Kang et al. 2021). They chose to refer to this region as DMPFC. They should discuss this issue further than just a footnote as other studies have been addressing the questions of the role of the DMPFC in social cognition. However, these other studies were stimulating a different brain area (Y=+31.5, Y=+44) (Ferrari et al. 2016 CABN; Wittmann et al. 2021 Neuron). Could the chosen ROI also be referred to as preSMA (See Johansen-Berg et al. 2004 PNAS)?

The authors compared the effect of a DMPFC stimulation with the effect of a caudal stimulation located at the junction of the pre and post central sulcus on the midline.

Some authors have reported sex differences in responses to social information (Li et al. 2020 Frontiers in

Human Neuroscience). Was there an effect of sex in the present study?

Does the protocol allow for detecting changes of performance over time? And for dissociating the impact of positive vs negative social outcomes?

Reviewer #3 (Remarks to the Author):

The paper assesses the effect of cTBS over dmPFC in an observational learning task with 2 conditions: action only observation vs. action-outcome observation in a demonstrator and quality of the decisions of the demonstrator (good vs bad). Model-free results suggest a fairly specific involvement of dmPFC in action-only observational learning when observing the good demonstrator. This effect pertains to both own choice and the predictions of the demonstrator's next choice. These findings are generally supported by computational reinforcement learning models that update an expectation signal with a weighted prediction error and an additional parameter modeling the modulatory effect of cTBS stimulation. An additional effect only shown with computational modeling is that dmPFC also affects the learning the predictability of the demonstrator, which is estimated as $1 - \text{entropy of the demonstrator's action probabilities}$.

The paper is clearly written with a specific question to answer equivocal accounts of the dmPFC involvement during observational learning. The TMS procedure follows established guidelines and the analyses are straightforward. I am a bit underwhelmed by the computational modeling as they are using numerous variations of a basic Rescorla-Wagner learning model targeting the value of different components of the task (more details below). The predictability model was computed using the approach by Charpentier et al., 2020, but was then also put under a RW learning rule. As it stands, these various RW learning models can iteratively approximate the findings uncovered in the model-free analyses and in my opinion offer little additional insights. Even the predictability model is primarily influenced by the experimental manipulation of the quality of the demonstrator: the entropy of a randomly choosing demonstrator is maximal and higher than for the good demonstrator and hence the predictability of both demonstrators differ in principle due to the experimental design. Overall, while I find the study interesting with a specific finding, but more suited for a more specific journal.

Detailed comments

Model recovery

It is laudable that the authors conducted a parameter recovery study and posterior predictive check as quality measures of model fitting, but in the context of the highly similar RW models a model recovery study is indispensable: can the model that generated synthetic data also be recovered from fitting this synthetic data with all model variants? This demonstration is crucial if the claim that these models target specific processes or task components is to be upheld.

Value ranges of TMS parameter (T_{choice} , $T_{\text{prediction}}$)

Inspect the scatter plots for the TMS parameter (T_{choice} , $T_{\text{prediction}}$) with the Probability differences

between stimulation site reveals that for T_{choice} (Fig 4 and S6). the bulk of the estimates parameter lie between 0.5 and 1, where as for $T_{prediction}$ the majority of value lie between 1 and 1.5. What is the reason for this and what effect does this have on the learning models. These TMS parameters act as a modulator of the learning rate-weighted prediction errors, so for the choice models the update term is further reduced leading to a slower learning of the target value, whereas for for the prediction models the effect of the update term is exaggerated. In fact, it might be that the combined effect of $T_{prediction}$ and learning rate might exceed the normal 0-1 range and hence accelerate rate the learning process to a degree lead to undamped increases in value estimate. To check for this state of affairs, the correlation between learning rate and the respective $T_{choice}/T_{prediction}$ parameter should be included in the supplement. Fig 4 show the presence of an outlier, so this correlation analysis should be repeated using robust regression (which might actually improve the finding). Also, the paper is lacking information about the posterior distribution of the fitted parameter values for the winning model.

Minor point

It would help of the enumeration of the models in the model comparison figures (Fig 3 and S5) would follow the identical order.

Page 23, line 6: The Predictability-Action model in Fig 3B is actually model 4 (not 5)

Fig.2: It might be worth considering choosing colors that are amenable for color-blind persons

Rebuttal

Reviewers' comments:

Reviewer #1 (Remarks to the Author):

Kang and colleagues report an interesting investigation of social learning using brain stimulation. They utilise a familiar paradigm that has been used by the authors and others in the past. They combine it with an offline brain stimulation protocol that disrupts dorsomedial prefrontal cortex (dmpfc) activity for a short period of time. The authors find stimulation effects on two outcome measures. After stimulation, participants predict other's choices less well and they are less well able to pick a higher-value option that they have to infer from others' choices. Both findings indicate that dmpfc stimulation disrupts learning from others and the impairments only occur when there is actually a relationship between the demonstrator's choices and the outcomes (i.e., only for a superb, not a random demonstrator).

In my view, the study is well done. Even though the effects of interest are not significant in omnibus tests, they are significant when directly comparing between the stimulation condition and a vertex control condition, which is critical. The tests are well justified by a priori hypotheses. Personally, I don't think the RL modelling and in particular the idea that dmpfc stimulation specifically affects "predictability" adds very much. That effect seems to occur because learning is only affected for the superb but not the random demonstrator – so only in situation where learning from the demonstrator makes sense. Nevertheless, methods wise, the RL modelling is very well done and thoroughly conducted using advanced fitting techniques, which is appropriate given the small number of trials per participants (due to the brain stimulation time restrictions). Also, the use of a non-social control condition is a strength of the paper.

We thank the reviewer for the positive evaluation of our study! We have carefully addressed the issues that were raised, which enhanced the clarity of our manuscript.

Major points:

1. I strongly urge the authors to show a figure with the brain stimulation coordinates overlaid on a structural image of the brain in main paper. This is because the authors stimulate a region of the brain that is usually not referred to as "dorsomedial prefrontal cortex". It is a very posterior region close pre-SMA. The authors note that they refer to this region as dmpfc, because the paper they take the coordinates from talks of dmpfc. This is legitimate, but it is also necessary to acknowledge that rarely anyone else refers to this region as dmpfc. That is why I think there should be a figure showing clearly where in the brain the brain stimulation took place.

Thank you for the helpful comment. We have now included a figure in the main paper that illustrates the stimulation site based on its MNI coordinates. We believe that the figure helps clarify the precise location of the brain stimulation in our study and hope that it reflects what the reviewer had in mind.

With respect to the anatomical labelling of the site we stimulated, we agree that this is a very complicated issue. We have searched multiple widely used neuroanatomical atlases to find the most appropriate label. On the one hand, the probabilistic cytoarchitectonic map in the Anatomy toolbox of

SPM (JuBrain) designates 52.4% of this area as preSMA, with its anterior border neighboring an uncharted part of the superior frontal gyrus. On the other hand, in the (non-probabilistic) EBRAINS, the same coordinate, based on the same parcellation, is categorized as already lying in this uncharted part, just anterior of pre-SMA. Given this ambiguity, we have decided to adopt a more conservative approach in the manuscript. We now refer to the area as DMPFC/preSMA throughout. This adjustment more clearly reflects the complexity of accurately labeling the stimulation site.

Probabilistic cytoarchitectonic maps (by Anatomy toolbox in SPM)

Multilevel Human Brain Atlas of the HBP on EBRAINS (<https://atlases.ebrains.eu/viewer>)

Additionally, we added a paragraph to discuss the definition of this brain region.

On p20,

“It is noteworthy that there is ambiguity when it comes to the precise anatomical labelling of the site stimulated, as the term “DMPFC” refers to a wide brain region in superior frontal gyrus. Our stimulation site was based on coordinates from a previous study (Suzuki et al., 2012); it is located at the border

between Brodmann areas 6 and 9 in a posterior part of mPFC. The terminology used in other studies for this area ranges from preSMA (Charpentier et al., 2020) to DMPFC (Ferrari et al., 2016, Kang et al., 2021, Suzuki et al., 2012), posterior mPFC, and dorsal anterior cingulate cortex/rostral cingulate zone (Izuma et al., 2011, Izuma et al., 2015, Klucharev et al., 2009, Klucharev et al., 2011). Due to the complexity of a precise definition, we opted to refer to the stimulation site as “DMPFC/preSMA”, indicating that assignment of a single and precise anatomical label is difficult.”

2. Model comparisons: The models are sufficiently described, but I had a very hard time figuring out how the model comparisons were conducted and in particular which data the models were fitted on. It was unclear which of the two decisions the data was fitted on (or both at the same time?). Also it was unclear if the whole sample was fitted together or not. The statement on page 12 “model comparison indicated that the Predictability-Action model (model 5) provided the best account for the behavioural effect of DMPFC cTBS” sounds like the model was specifically fitted on the differences between conditions?

Thank you for your insightful comment. We apologize for any confusion in describing how the models were fitted. To clarify, we conducted two separate model fits, one for the "choice for self" and the other for "prediction of demonstrator actions". For each of these two types of decisions, all the social trials (i.e., all the observational learning conditions without the individual learning condition) were fitted together which we now specify more clearly in the captions of Figure 3 and Figure S5 (please see below for wording).

The reason behind fitting "choice for self" and "prediction of demonstrator actions" separately was to avoid comparison of an overly extensive number of variations in the RW model. If we were to fit both decisions together, it would lead to additional variations, such as whether or not to assume the same parameters for both decisions, resulting in an overwhelming number of possible parameter combinations. To make this point clearer, we have added the following text to the manuscript:

On p 12 : “We fitted choice for self and prediction of demonstrator actions separately (to keep parameter space manageable) and attempted to find the best model for each behavior.”

“Figure 3. Models and model comparison for choice for self in the decision phase. A) A schematic graphic of the compared models. B) Model comparison for choice for self. Lower DIC values indicate better model fit. The model with predictability controlling action-based learning explained choice for self best.”

“Importantly, model comparison for choice for self data from all observational conditions indicated that the Predictability-Action model (model 5) provided the best account for the behavioral pattern across vertex and DMPFC/preSMA cTBS (Figure 3B).”

In SI,

“Figure S5. Model comparison for prediction of demonstrator action. Lower DIC values indicate better model fit. The model where DMPFC/preSMA downregulation affected predictability learning in the Action-Only condition explained prediction of demonstrator action best.”

To improve clarity, we also have reorganized the paragraph by first introducing components of the computational model that are common for both types of decision and then explaining the specific parts of the models for each type of decision.

On p 11,

“Computational model for choice for self and prediction of demonstrator actions

The model-independent analyses demonstrated that DMPFC/preSMA cTBS decreased both observational learning (choice for self) and the prediction of demonstrator actions”

On p12,

“DMPFC/preSMA downregulation disrupts learning about demonstrator predictability for choice for self

Importantly, model comparison for choice for self data from all observational conditions indicated”

On p16,

“DMPFC/preSMA cTBS disrupts predictability learning for prediction of demonstrator actions

To understand whether cTBS has similar effects on the choices of participants for themselves and their predictions of demonstrator choices, we fitted the models that we used to explain choice for self also to prediction of demonstrator actions in all observational conditions.”

3. Other recent accounts of learning about other people suggest that self-other-distinction is an issue (Ereira et al., 2018, 2020) and implicate specifically dmPFC in this ability, also using brain stimulation evidence (Wittmann et al., 2016, 2021). Could the authors findings alternatively also be explained by failures of self-other-distinction induced by brain stimulation? This could be tested in the model comparison or at the very least discussed as another possibility.

Thank you for the valuable comment and for pointing us to an interesting line of research! To address the possibility that self-other distinction may have played a role in our study, we checked the individual learning condition (learning from own outcome) and found no significant effect of stimulation. In other words, the stimulation effects were specific to learning from observing demonstrator actions, making it unlikely that self-other merge due to stimulation would lead to effects that are specific to just one condition. Please note though, because we stimulated a more posterior site, it is well possible that both these functions are governed by different portions of the DMPFC/pre-SMA. We elaborate on this issue in the Discussion and link our findings to other relevant studies.

“Some studies have demonstrated that the MPFC contributes to the distinction between self and other (Ereira et al., 2020, Wittmann et al., 2016), and that downregulation of the DMPFC leads to self-other merge when evaluating the abilities of oneself and other people (Wittmann et al., 2021). Our study focused on learning from observing the actions of others, but also formally tested if DMPFC stimulation changes learning from outcomes for oneself. We find that learning from own outcomes (individual learning condition) is not affected by DMPFC/preSMA cTBS, suggesting that the stimulation effects are specific to learning from observing (the actions of) others. Thus, possible merging of (the outcome representations of) self and other as a consequence of stimulation is unlikely to have caused the effects on observational learning in the present study. However, our study targeted a more posterior part of DMPFC compared to the studies of self-other merge (Wittmann et al., 2016, 2021). Thus, it is well possible that different portions of the DMPFC govern both functions. Moreover, the precise area in dorsal and ventral MPFC distinguishing self from other may depend on individual traits (Sul et al., 2015, Piva et al., 2019).”

Minor points

1. I strongly suggest to give a better intuition how the models work in the main text. They appear very obscure at the moment.

Thank you very much for your feedback! We added a description about the model in the beginning of the result as below.

P 11,

“Rather than directly learning from the actions or outcomes of the demonstrator, the models in the second group (Models 4-6) tested whether learning about the predictability of the demonstrators was affected by DMPFC/preSMA cTBS. According to the models in this group, learning takes place in two stages, 1) learning about the predictability of demonstrator actions 2) learning from outcome and action prediction errors, with the degree of learning modulated by 1). Models 4-6 commonly assumed that DMPFC/preSMA cTBS affected predictability learning, but differed in whether the learned demonstrator predictability affected only observational outcome learning (Model 4, Predictability-Outcome model), only observational action learning (Model 5, Predictability-Action model) or both forms of learning (Model 6, Predictability-both model). In these models, the predictability of demonstrators could be assessed through an internal simulation of demonstrator behavior. By simulating observed behavior over trials, the observers learned to what degree demonstrator choices became more predictable (by accumulating the observed choice predictability, see Methods), which modulated action- and/or outcome-based learning in the models of the second group. The first (Models 1-3) and the second (Models 4-6) group of models can have similar predictions at the group level, but the underlying cognitive processes are different.”

2. When I read the manuscript, I kept getting confused about which of the two decisions that are in a trial the authors talked about – the demonstrator-prediction or the individual-stimulus-choice. I suggest signposting this better throughout the manuscript and not just talking about “the choice” in parts of the results section.

We appreciate this suggestion by the Reviewer and now use the wording “choice for self” (instead of “choice”) and “prediction of demonstrator actions” (instead of “prediction”) throughout the manuscript.

3. A table displaying the free parameters per model could be useful to get an overview over the models.

Thank you very much for the valuable comment. We added a table of the free parameters per model in the supplement information.

Models		Parameters	Parameters affected by τ (DMPFC/preSMA TMS)
Model 1	Outcome learning decreased	$\alpha, \beta, \kappa, w, \rho, \tau$	α
Model 2	Action learning decreased	$\alpha, \beta, \kappa, w, \rho, \tau$	κ
Model 3	Action-outcome learning decreased	$\alpha, \beta, \kappa, w, \rho, \tau$	α, κ
Model 4	Predictability: Outcome learning	$\alpha, \beta, v, w, \rho, \tau$	v (affecting outcome based learning)
Model 5	Predictability: Action learning	$\kappa, \beta, v, w, \rho, \tau$	v (affecting action based learning)
Model 6	Predictability: Action and outcome learning	v, β, w, ρ, τ	v (affecting action and outcome based learning)

4. Page 10/line 10: “The results remained qualitatively the same when controlling for stimulation order.” – Yes, but were the effects of interest still significant?

Thank you very much for the comment. All the results remained significant. Below, we have added them in brackets. In the manuscript, we mention more generically that “*all effects remained qualitatively the same and significant or non-significant when controlling for stimulation order*” (p7; we repeat a similar statement in the Methods section, p27) in order to increase readability. However, we would be happy to add them to the text, if the Reviewer prefers.

Results from p7

Predictably, we found better performance when participants observed the superb demonstrator than when they observed the bad demonstrator, $b = -0.487$, $z = -2.22$, $p = .026$ ($z = -2.22$, $p = .026$), Bayes factor₁₀ = 5.08. Moreover, demonstrator quality interacted with observed information, $b = -0.524$, $z = -2.12$, $p = .034$ ($z = -2.12$, $p = 0.03$), Bayes factor₁₀ = 3.35.

Results from p8

DMPFC/preSMA downregulation compared to control (vertex) stimulation decreased performance in the Action-Only condition with the superb demonstrator, $b = 0.341$, $z = 2.54$, $p = .011$, ($z = 2.521$, $p = .011$) Bayes factor₁₀ = 7.26. In contrast, Action-Outcome learning from the bad demonstrator remained unimpaired, $b = 0.017$, $z = 0.078$, $p = 0.938$ ($z = 0.07$, $p = .941$), Bayes factor₁₀ = 0.022, (Figure 2A). Thus, DMPFC/preSMA cTBS appeared to decrease action-based observational learning rather selectively. We also examined if DMPFC/preSMA downregulation affected Action-Outcome learning with superb demonstrators, and found that DMPFC/preSMA downregulation had no significant effect, $b = 0.131$, $z = 0.517$, $p = 0.605$ ($z = 0.522$, $p = 0.601$), Bayes factor₁₀ = 0.066. Thus, it is possible that outcome-based learning compensated for the decrease in action-based learning. Lastly, no difference was found in Action-Only learning from bad demonstrators, $b = 0.1533$, $z = 0.760$, $p = 0.447$ ($z = 0.760$, $p = .447$), Bayes factor₁₀ = 0.051.

Purely individual learning performance was also unaffected by DMPFC/preSMA cTBS, $b = -0.056$, $z = -0.285$, $p = 0.776$ ($z = -0.285$, $p = 0.776$) Bayes factor₁₀ = 0.029.

Results from p9

A generalized linear mixed model revealed, as expected, that participants predicted the choices of the superb demonstrator more accurately than those of the bad demonstrator, $b = -1.44$, $z = -9.78$, $p < .001$ ($z = -9.17$, $p < .001$), Bayes factor₁₀ = 1.94×10^8 . Observing both actions and outcomes showed a (non-significant) trend-level increase of prediction performance compared to observing only actions, $b = -.248$, $z = -1.73$, $p = .084$ ($z = -1.72$, $p = .084$), Bayes factor₁₀ = 0.578.

Results from p10

These revealed that compared to vertex stimulation, DMPFC/preSMA downregulation resulted in poorer prediction performance with the superb demonstrator in the Action-Only condition, $b = 0.264$, $z = 2.368$, $p = .018$ ($z = 2.376$, $p = .0175$), Bayes factor₁₀ = 0.78, but neither in the Action-Outcome condition, $b = 0.169$, $z = 1.373$, $p = 0.169$ ($z = 1.367$, $p = 0.171$), Bayes factor₁₀ = 0.078, nor in the Action-Outcome learning condition with the bad demonstrator, $b = 0.067$, $z = -0.681$, $p = 0.469$ ($z = -0.684$, $p = .493$), Bayes factor₁₀ = 0.031; **Figure 2B**).

5. The section about “imitation” on page 10. I think the point here is that brain stimulation effects cannot be explained as changing people’s tendency to blindly imitate choices. If that is true, could the authors simply state that? I found the long explanation of what choice history is a bit misplaced and felt it did not help the authors make their point. Of course the authors may also keep it as is and try to make clearer what they mean.

Thank you very much for the comment. To make the point clearer, we fitted an additional model which

is a variation of model 2 testing for imitation behavior, by fitting a weight parameter for choosing the same behavior as the demonstrator by changing Eq 4 as below. This part is included in the SI:

“In order to examine whether DMPFC/preSMA cTBS affected imitation behavior instead of action-based learning, we fitted a variation of model 2 to all choice for self data in observational conditions. By changing Eq 4 as detailed below, we investigated whether the tendency of choosing the option chosen by a demonstrator is influenced by DMPFC/preSMA cTBS. The model fit was 7349 which is worse than that of models 1-6. “

Accordingly, we concluded that DMPFC/preSMA cTBS did not primarily affect imitation of demonstrator behavior.

$$\text{Eq 4.} \quad AP(a)_i = AP(a)_i + \kappa * (A_i^{\text{demonstrator}} - AP(a)_i)$$

$$\text{Imitation-V} \quad AP(a)_i = \text{weighting of imitation} * A_i^{\text{demonstrator}}$$

$$\text{Imitation-D} \quad AP(a)_i = \tau * \text{weighting of imitation} * A_i^{\text{demonstrator}}$$

$$A_i^{\text{demonstrator}} \begin{cases} \text{option chosen by a demonstrator} : 1 \\ \text{if not} : 0 \end{cases}$$

Also, we modified the manuscript as below.

On p 10,

“Imitation. Given that choice and prediction performance in the Action-Only condition were impaired by DMPFC/preSMA downregulation, one may ask if DMPFC/preSMA downregulation affected the ability of our participants to imitate (i.e., to select exactly the same options as the demonstrator in each trial) rather than the ability to learn from observed actions. We differentiated between action-based learning and imitation by examining whether the accumulation of knowledge across multiple trials was necessary or not. A generalized linear mixed model analyzing imitation behavior (see Methods) revealed no difference between vertex and DMPFC/preSMA downregulation in either condition (Figure S2). Moreover, planned pairwise comparisons between stimulation sites showed no difference in imitation between vertex and DMPFC/preSMA downregulation. In addition, we examined a model with imitation behavior instead of action-based learning, and it fit the data worse than most other models (SI). Thus, we find little support for stimulation effects on imitation. This means that the DMPFC/preSMA cTBS effect in the Action-Only condition with the superb demonstrator did not result from copying the demonstrator’s choice in each trial, but from observational action learning. “

Reviewer #2 (Remarks to the Author):

In this study, Kang and colleagues are investigating the impact of a repeated TMS stimulation of a structure of the dorsomedial prefrontal cortex in observational learning.

The authors chose a stimulation site based on a previous study by Suzuki and colleagues. The chosen site is also close to the cluster they identified in their previous study (Kang et al. 2021). They chose to refer to this region as DMPFC. They should discuss this issue further than just a footnote as other studies have been addressing the questions of the role of the DMPFC in social cognition. However, these other studies were stimulating a different brain area (Y=+31.5, Y=+44) (Ferrari et al. 2016 CABN; Wittmann et al. 2021 Neuron). Could the chosen ROI also be referred to as preSMA (See Johansen-Berg et al. 2004 PNAS)?

Thank you very much for raising this important point. We originally defined this area as DMPFC based on the Suzuki et al., (2012) paper instead of our paper, Kang et al. (2021), as we planned this project before our paper was published. We have now included a figure in the main paper that illustrates the stimulation site based on its MNI coordinates:

With respect to the anatomical labelling of the site we stimulated, we agree that this is a very complicated issue. We have searched multiple widely used neuroanatomical atlases to find the most appropriate label. On the one hand, the probabilistic cytoarchitectonic map in the Anatomy toolbox of SPM (JuBrain) designates 52.4% of this area as preSMA, with its anterior border neighboring an uncharted part of the superior frontal gyrus. On the other hand, in the (non-probabilistic) EBRAINS, the same coordinate, based on the same parcellation, is categorized as already lying in this uncharted part, just anterior of pre-SMA. Given this ambiguity, we have decided to adopt a more conservative approach in the manuscript. We now refer to the area as DMPFC/preSMA throughout. This adjustment more clearly reflects the complexity of accurately labeling the stimulation site.

Probabilistic cytoarchitectonic maps (by Anatomy toolbox in SPM)

Multilevel Human Brain Atlas of the HBP on EBRAINS

(<https://atlases.ebrains.eu/viewer>)

Additionally, we added a paragraph to discuss the definition of this brain region.

On p20,

“It is noteworthy that there is ambiguity when it comes to the precise anatomical labelling of the site we stimulated, as the term “DMPFC” refers to a wide brain region in superior frontal gyrus. Our stimulation site was based on coordinates from a previous study (Suzuki et al., 2012); it is located at the border between Brodmann areas 6 and 9 in a posterior part of mPFC. The terminology used in other studies for this area ranges from preSMA (Charpentier et al., 2020) to DMPFC (Ferrari et al., 2016, Kang et al., 2021, Suzuki et al., 2012), posterior mPFC, and dorsal anterior cingulate cortex/rostral cingulate zone (Izuma et al., 2011, Izuma et al., 2015, Klucharev et al., 2009, Klucharev et al., 2011). Due to the complexity of a precise definition, we opted to refer to the stimulation site as “DMPFC/preSMA”, indicating that assignment of a single and precise anatomical label is difficult.”

The authors compared the effect of a DMPFC stimulation with the effect of a caudal stimulation located at the junction of the pre and post central sulcus on the midline.

Some authors have reported sex differences in responses to social information (Li et al. 2020 *Frontiers in Human Neuroscience*). Was there an effect of sex in the present study?

Thank you very much for the comment. We performed general linear model analyses to investigate the effect of sex but found little evidence ($b = 0.10$, $z = -0.07$, $p = 0.50$) in choice for self and prediction of demonstrator actions ($b = 0.02$, $z = 0.40$, $p = 0.69$). We included this in the manuscript as below.

On p8

“Moreover, there was no effect of sex ($b = 0.10$, $z = -0.07$, $p = 0.50$).”

On p10

“Moreover, there was no effect of sex ($b = 0.02$, $z = 0.40$, $p = 0.69$).”

Does the protocol allow for detecting changes of performance over time?

Thank you for your valuable feedback! Indeed, the task design allows us to detect changes in

performance over time. We tested GLMMs separately for each condition and stimulation site, assessed the effect of trial number to operationalize changes over time, and tested whether trial number interacted with stimulation site. In particular, in the Action-Only condition with the superb demonstrator, the effect of trial was significant in choice for self $z = 10.73$, $p < 0.001$ and in prediction of demonstrator actions, $z = 8.44$, $p < 0.001$. Thus, the protocol does indeed allow for detecting changes of performance over time. However, there was no interaction with stimulation site, either for choice for self $z = 0.55$, $p = 0.58$, or for prediction of demonstrator actions, $z = -0.87$, $p = 0.38$ in the Action-Only condition with the superb demonstrator. Thus, our primary findings were unaffected by learning.

And for dissociating the impact of positive vs negative social outcomes?

Thank you for raising this important point. In our task, the demonstrator's outcomes were either 0 or 10, and as such, there were no negative outcomes. However, the alignment or misalignment with expectations in action-based learning can be considered as social outcomes. To explore the impact of positive vs. negative social outcomes, we examined whether having separate learning rates for positive and negative social action prediction errors improves the fit of the best model by modifying Eq 9:

$$\text{If } (A_i^{\text{demonstrator}} - AP(a)_i) > 0, \quad AP(a)_i = AP(a)_i + AcP_i * \kappa_{\text{pos}}(A_i^{\text{demonstrator}} - AP(a)_i)$$

$$\text{If } (A_i^{\text{demonstrator}} - AP(a)_i) \leq 0, \quad AP(a)_i = AP(a)_i + AcP_i * \kappa_{\text{neg}}(A_i^{\text{demonstrator}} - AP(a)_i)$$

After model fitting, we found that the model with separate learning rates for positive and negative social action prediction errors did not improve the overall model fit (DIC of Model 5: 6292, DIC of model with separate learning rates for positive and negative prediction errors: 6635). Thus, there was no advantage in dissociating the impact of positive and negative social outcomes in our study. For conciseness, we have not included this information in the manuscript but would be happy to do so if you would like us to.

Reviewer #3 (Remarks to the Author):

The paper assesses the effect of cTBS over dmPFC in an observational learning task with 2 conditions: action only observation vs. action-outcome observation in a demonstrator and quality of the decisions of the demonstrator (good vs bad). Model-free results suggest a fairly specific involvement of dmPFC in action-only observational learning when observing the good demonstrator. This effect pertains to both own choice and the predictions of the demonstrator's next choice. These findings are generally supported by computational reinforcement learning models that update an expectation signal with a weighted prediction error and an additional parameter modeling the modulatory effect of cTBS stimulation. An additional effect only shown with computational modeling is that dmPFC also affects the learning the predictability of the demonstrator, which is estimated as 1 - entropy of the demonstrator's action probabilities.

The paper is clearly written with a specific question to answer equivocal accounts of the dmPFC involvement during observational learning. The TMS procedure follows established guidelines and the analyses are straightforward. I am a bit underwhelmed by the computational modeling as they are using numerous variations of a basic Rescorla-Wagner learning model targeting the

value of different components of the task (more details below). The predictability model was computed using the approach by Charpentier et al., 2020, but was then also put under a RW learning rule. As it stands, these various RW learning models can iteratively approximate the findings uncovered in the model-free analyses and in my opinion offer little additional insights. Even the predictability model is primarily influenced by the experimental manipulation of the quality of the demonstrator: the entropy of a randomly choosing demonstrator is maximal and higher than for the good demonstrator and hence the predictability of both demonstrators differ in principle due to the experimental design. Overall, while I find the study interesting with a specific finding, but more suited for more specific journal.

We thank the reviewer for finding our paper “clearly written with a specific question to answer equivocal accounts of the dmPFC involvement during observational learning”. We also are grateful for the constructive feedback for the study. Regarding modeling, it is important to note that our paper uses computational modeling with the aim to pinpoint the specific latent process implemented by the DMPFC during observational learning and that this aim of modeling requires systematic variation within a common model structure. Accordingly, it would not make sense if one were to compare widely different and novel models because this would not answer the specific question we set out to answer and would not allow us to compare our findings with previous modeling approaches using variants of our paradigm (Burke et al., 2010, Kang et al., 2021). Our findings substantially advance the model-related insights gained from these studies by showing that extensions of a Rescorla-Wagner model can account for demonstrator quality and by revealing which exact processes DMPFC stimulation causally affects.

We now mention these points on p. 11 of the main manuscript:

“Pinpointing the specific latent process implemented by the DMPFC during observational learning with computational modeling requires systematic model variation and comparison within a common model structure, in keeping with the notion that computational modeling needs to be tailored to the diverse purposes it can serve (Kording et al., 2018). Accordingly, rather than comparing widely different and novel models, what is needed is to build upon, and systematically advance, previous modeling approaches (Burke et al., 2010, Kang et al., 2021) that were based on a closely related behavioral paradigm.”

Detailed comments

Model recovery

It is laudable that the authors conducted a parameter recovery study and posterior predictive check as quality measures of model fitting, but in the context of the highly similar RW models a model recovery study is indispensable: can the model that generated synthetic data also be recovered from fitting this synthetic data with all model variants? This demonstration is crucial if the claim that these model target specific processes or task components is to be upheld.

Thank you for your insightful comment. We fully agree that it is important to conduct model recovery, especially if models are similar. We have now included the model recovery results for both "choice for self" and "prediction of demonstrator action" separately in the SI. As demonstrated in the confusion matrix below, the model for "choice for self" is perfectly identifiable, which strengthens the validity of our approach. Additionally, the models for "prediction of demonstrator action" were identifiable in the "Predictability learning" model group, further supporting the specificity of the models targeting distinct processes or task components. Thus, the models indeed target specific processes.

We now mention these points in the main manuscript:

On p35

“We also performed model recovery for each model to verify whether the models are distinguishable given the data and the fitting procedure. We ran 50 simulations of each model with the data of each participant and each trial. For the simulation, model parameters were randomly chosen within the range of the parameters. After simulation, we fitted models on the data generated by each model and report the confusion matrix (Figure S10). The model recovery confirmed that the models were identifiable for choice for self and for models learning predictability in order to predict demonstrator actions.”

Model recovery of models for Choice for self

Fit model

	Model 1	Model 2	Model 3	Model 4	Model 5	Model 6
Simulated model	Model 1	0.82	0.1	0.08	0	0
Model 2	0	0.98	0.02	0	0	0
Model 3	0	0.06	0.94	0	0	0
Model 4	0	0	0	1	0	0
Model 5	0	0	0	0	1	0
Model 6	0	0	0	0	0	1

Model recovery of models for Prediction of demonstrator action

Fit model

	Model 1	Model 2	Model 3	Model 4	Model 5	Model 6	
Simulated model	Model 1	0.36	0.04	0.02	0.08	0.46	0.04
Model 2	0.16	0.16	0.02	0.12	0.5	0.04	
Model 3	0.24	0.18	0.02	0.14	0.38	0.04	
Model 4	0	0	0	1	0	0	
Model 5	0	0	0	0	0.92	0.08	
Model 6	0	0	0	0	0	1	

Value ranges of TMS parameter (Tchoice, Tprediction)

Inspection the scatter plots for the TMS parameter (Tchoice, Tprediction) with the Probability differences between stimulation site reveals that for Tchoice (Fig 4 and S6). the bulk of the estimates parameter lie between 0.5 and 1, where as for Tprediction the majority of value lie between 1 and 1.5. What is the reason for this and what effect does this have on the learning models. These TMS parameters act as a modulator of the learning rate-weighted prediction errors, so for the choice models the update term is further reduced leading to a slower learning of the target value, whereas for for the prediction models the effect of the update term is exaggerated. In fact, it might be that the combined effect of Tprediction and learning rate might exceed the normal 0-1 range and hence accelerate rate the learning process to a degree lead to undamped increases in value estimate. To check for this state of affairs, the correlation between learning rate and the respective Tchoice/Tprediciton parameter should be included in the supplement. Fig 4 show the presence of an outlier, so this correlation analysis should be repeated using robust regression (which might actually improve the finding). Also, the paper is

lacking information about the posterior distribution of the fitted parameter values for the winning model.

Thank you very much for improving understandability of the model. We have now performed robust regression analyses and confirmed that there were no correlations between the predictability learning parameter and either T_{choice} (coefficient = 0.06, $p=0.25$), or $T_{\text{prediction}}$ (coefficient = 0.004, $p = 0.82$). Also, we have checked whether the multiplication of learning weight by T_{choice} and $T_{\text{prediction}}$ exceeds 1 (which could result in excessive learning). We found that the means of u parameters (predictability learning) were 0.06 ± 0.018 in both “choice for self” and “prediction of demonstrator action”, which results in a range for the learning rate weighted by T_{choice} and $T_{\text{prediction}}$ well within the normal 0-1 range. Thus, there was no excessive updating in the learning models. We have added this information to the manuscript.

On p34, we write:

“To test whether the learning rate u and T_{choice} are correlated, we performed robust regression analyses and found no significant correlation between u and either T_{choice} (coefficient = 0.06, $p=0.25$) or $T_{\text{prediction}}$ (coefficient = 0.004, $p = 0.82$). In addition, the multiplied values of the two parameters were within the normal range for learning processes (0-1), indicating that there was no excess in the predictability learning process or in the resulting value estimate.”

Based on the Reviewer’s comment, we realized that the original plot depicting the distribution of $T_{\text{prediction}}$ actually came from a model with a different prior. We sincerely appreciate that the reviewer noticed this mistake. $T_{\text{prediction}}$ from the model we actually used in the rest of the manuscript distributed mostly below 1 as shown in the corrected plot (below and figure S6).

on p 17

“While the correlation was in the expected direction, it was not statistically significant, Spearman’s rho = -0.25, $p = 0.10$, one-tailed (Figure S6). However, the corresponding correlation with the difference in the proportion of correct choice for self was statistically significant, Spearman’s rho = -0.31, $p = 0.04$, one-tailed, Bayes factor = $1.80 \cdot 10^{37}$ (Figure S6).”

We also added the information about the posterior distribution of the fitted parameters in SI.

Figure S11. Posterior distributions of the parameters of Model 5 for A) choice for self and B) prediction of demonstrator action.

Minor point

It would help if the enumeration of the models in the model comparison figures (Fig 3 and S5), would follow the identical order.

Thank you for the comment. We have changed Figure 3 and S5 according to the order of the models in the text.

Figure 3

Figure S5

Page 23, line 6: The Predictability-Action model in Fig 3B is actually model 4 (not 5)

Thank you for bringing this to our attention. We apologize for the oversight. We have now corrected the figure and made sure that the corresponding text is consistent with the figure.

Fig.2: It might be worth considering choosing colors that are amenable for color-blind persons

We sincerely appreciate this comment. We have changed the colors of the graphs to be more accessible for color-blind people.

REVIEWER COMMENTS

Reviewer #1 (Remarks to the Author):

The authors have addressed my concerns very well. The manuscript has been very good before already. The main results are very clear. Stimulation over dmPFC/pre-SMA influences how we learn about other people. This finding is exciting and an important addition to social neuroscience and brain stimulation research. I expect that the paper will find the broad audience that it deserves in Nature Communications.

The authors have comprehensively addressed my concern about the anatomical location of the stimulation site by including a figure showing the stimulation sites in the main manuscript, and by adjusting the label for the stimulated region (dmPFC/pre-SMA). It is now very clear to the reader where neural activity was changed using TMS. My clarification questions about the computational model were also clearly addressed, even more thoroughly than needed. The statistical questions were well answered and further demonstrate remarkably strong effects of the brain stimulation on social learning. Also, the issue about whether the learning effects were driven by changes in imitation behaviour alone was tackled thoroughly including a new variant of the RL model to rule out this alternative explanation. In sum, the authors report an interesting and novel investigation showing the causal importance of dmPFC/pre-SMA for observational learning. Many people will be interested in this result, and I think it fully deserves to be published in Nature Communications.

Reviewer #3 (Remarks to the Author):

I appreciate the work that the authors have undertaken to address my comments. The model recovery analysis is convincing and the plots of the posterior parameter distribution are informative, but also troublesome, which is part of the reason why I am still unconvinced that this manuscript has really grown into a Nature Communications paper.

Learning rates and softmax temperatures for the model of the demonstrator remain arbitrarily fixed to "reduce the number of free parameters in the model". However, there are now quite a few approaches that put constraints on mental models of others through a hierarchical learning model (e.g. HGF - Behrens, Nature 2008; Mathys, Front Hum Neurosci, 2011; Diaconescu, PLoS CB, 2014). Testing a few other parameters around the fixed value (and applying them to all subjects identically) does not give credit to the variability of such parameters in a given sample. While I think that a

hierarchical *cognitive* Bayesian model (not just hierarchical estimation via jags/Stan/PyMC) is more appropriate for these data, at least a grid search to find the optimal parameters for the model of the other's decision process should have been warranted.

The posterior distribution of the model parameters are also partially troublesome. The peaking of the distributions for alpha, beta, and rho at the end of the pre-specified parameter ranges suggests that given the current model of self choice and demonstrator predictability the sampler would have drawn sample beyond the pre-determined range. This implies that there might a model misspecification going on here. Alternatively, there might be a strong correlation between learning rate and temperature at play, which is frequently observed with Rescorla-Wagner type learning models: given a certain experimental variability, the model can either capture this with a very high learning rate and a low temperature or the other way around. A careful discussion and interpretation of these posterior distribution is lacking in the paper.

In a broad viewer of the modeling efforts, I still think that it offers very little beyond the model-free analysis of the mixed-models approach. The way the predictability model is constructed just replicates the experimental conditions for the demonstrator. Predictability (1-entropy) is just a measure of how much the choice history of the demonstrator deviates from 50/50 and this is exactly the how the demonstrators are constructed in the experiment. Because the neural manipulation is done off-line, the additional insight that a correlation of brain activity with model-derived variables does not apply for this study (i.e. not functional images of brain activity were collected).

Reviewer #4 (Remarks to the Author):

Follow-up comment on the 1st comment by Reviewer #2:

Given the inherent difficulty of precisely specifying anatomical labels of the human brain normalized to a template (e.g., MNI and Talairach atlases), it is fair to describe the stimulation site as DMPFC/preSMA.

Follow-up comment on the last comment by Reviewer #2:

The description of the computational models could be significantly improved.

For example, it is difficult to understand because the intuitive explanation of the formula, the description of the symbols within it, and the formula itself are described in separate locations.

Moreover, the description of the Bayesian hierarchical model is unclear. In a Bayesian hierarchical model, each subject's individual-level parameter (e.g., learning rate) is assumed to be generated from the group-level distribution, such as a normal or beta distribution (its parameters are denoted as population-level parameters). See <https://www.princeton.edu/~ndaw/d10.pdf>. As far as I understand, the authors assume the individual-level parameter, a , is drawn from the group-level Beta distribution that has two parameters (say m and n). Then, what does the description “ $a \sim \text{dunif}(0.001, 0.999)$ ” mean? What is shown in Fig. A9A? The authors should provide a formal and comprehensive description.

REVIEWER COMMENTS

Reviewer #1 (Remarks to the Author):

Reviewer comment: The authors have addressed my concerns very well. The manuscript has been very good before already. The main results are very clear. Stimulation over dmPFC/pre-SMA influences how we learn about other people. This finding is exciting and an important addition to social neuroscience and brain stimulation research. I expect that the paper will find the broad audience that it deserves in Nature Communications.

The authors have comprehensively addressed my concern about the anatomical location of the stimulation site by including a figure showing the stimulation sites in the main manuscript, and by adjusting the label for the stimulated region (dmPFC/pre-SMA). It is now very clear to the reader where neural activity was changed using TMS. My clarification questions about the computational model were also clearly addressed, even more thoroughly than needed. The statistical questions were well answered and further demonstrate remarkably strong effects of the brain stimulation on social learning. Also, the issue about whether the learning effects were driven by changes in imitation behaviour alone was tackled thoroughly including a new variant of the RL model to rule out this alternative explanation. In sum, the authors report an interesting and novel investigation showing the causal importance of dmPFC/pre-SMA for observational learning. Many people will be interested in this result, and I think it fully deserves to be published in Nature Communications.

Response: We sincerely appreciate this positive evaluation. Your comments were invaluable to the refinement of this work. Thank you very much.

Reviewer #3 (Remarks to the Author):

Reviewer comment: I appreciate the work that the authors have undertaken to address my comments. The model recovery analysis is convincing and the plots of the posterior parameter distribution are informative, but also troublesome, which is part of the reason why I am still unconvinced that this manuscript has really grown into an Nature Communications paper. Learning rates and softmax temperatures for the model of the demonstrator remain arbitrarily fixed to "reduce the number of free parameters in the model". However, there are now quite a few approaches that put constraints on mental models of others through a hierarchical learning model (e.g. HGF - Behrens, Nature 2008; Mathys, Front Hum Neurosci, 2011; Diaconescu, PLoS CB, 2014). Testing a few other parameters around the fixed value (and applying them to all subjects identically) does not give credit to the variability of such parameters in a given sample. While I think that a hierarchical *cognitive* Bayesian model (not just hierarchical estimation via jags/Stan/PyMC) is more appropriate for these data, at least a grid search to find the optimal parameters for the model of the other's decision process should have been warranted.

Response: Thank you very much for your constructive comments! As you suggested, we now conducted a comprehensive grid search for the optimal parameter combinations. Specifically, we examined a range of demonstrator parameters, varying alpha from 0.1 to 1 in increments of 0.1 and beta from 1 to 10 in increments of 1. We did so for Models 4-6, but not for Models 1-3 for which the model fit was not affected by the alpha and beta of demonstrators. Please note that in response to your second comment, we have modified the model and the range of parameters we tested (see also below). Thus, the grid search was implemented for this revised model and the adjusted parameter ranges.

The results of the grid search are shown in New Table S5, and the results for Model 5 are further visualized in Table S6. As before, we find that 1) Model 5 is the best model across all parameter combinations, and 2) the model fit best for an alpha of 0.2 and a beta of 9 for demonstrators. We therefore report these parameter values and employ them for all further model fits and analyses.

Table S5. Model comparisons (DIC) of Model 4-6 to determine the optimal alpha and beta of demonstrators

Beta	Alpha	Model 4	Model 5	Model 6
1	0.1	6747.94	6521.257	6999.145
2	0.1	6603.942	6341.256	6671.202
3	0.1	6565.05	6240.218	6509.514
4	0.1	6541.75	6173.442	6412.207
5	0.1	6534.333	6149.626	6369.11
6	0.1	6515.738	6139.623	6366.919
7	0.1	6508.835	6101.817	6345.298
8	0.1	6515.875	6113.723	6345.246
9	0.1	6538.312	6114.926	6340.558
10	0.1	6547.382	6121.371	6358.262
1	0.2	6621.946	6344.984	6649.175
2	0.2	6564.507	6172.838	6432.373
3	0.2	6534.385	6133.271	6399.831
4	0.2	6544.654	6093.83	6389.639
5	0.2	6552.621	6105.8	6387.422
6	0.2	6560.638	6090.231	6392.113
7	0.2	6549.543	6094.046	6411.119
8	0.2	6558.948	6089.398	6404.292
9	0.2	6570.394	6086.845	6412.305
10	0.2	6546.346	6087.748	6427.336
1	0.3	6603.193	6247.839	6542.58
2	0.3	6573.331	6132.319	6438.524
3	0.3	6562.381	6109.627	6456.405
4	0.3	6579.105	6109.019	6464.626
5	0.3	6569.286	6098.276	6472.486
6	0.3	6573.051	6091.755	6478.319
7	0.3	6576.158	6099.483	6481.797
8	0.3	6570.271	6099.278	6506.675
9	0.3	6567.553	6094.741	6502.432
10	0.3	6568.03	6100.172	6491.372
1	0.4	6627.748	6220.095	6519.3
2	0.4	6622.202	6130.739	6541.699
3	0.4	6634.926	6128.296	6551.524
4	0.4	6655.349	6133.924	6559.684
5	0.4	6652.81	6121.55	6571.001
6	0.4	6667.795	6114.026	6555.085
7	0.4	6671.233	6140.604	6558.772
8	0.4	6654.218	6126.193	6568.321
9	0.4	6655.274	6120.563	6577.612
10	0.4	6653.001	6118.368	6569.763
1	0.5	6695.706	6239.265	6603.938
2	0.5	6712.873	6178.575	6627.308
3	0.5	6726.444	6162.717	6641.159
4	0.5	6709.941	6164.483	6640.158
5	0.5	6723.522	6144.326	6658.849
6	0.5	6717.57	6165.723	6665.559
7	0.5	6723.334	6160.01	6665.01
8	0.5	6724.677	6152.878	6654.504
9	0.5	6725.564	6145.676	6651.143
10	0.5	6701.796	6160.566	6649.439
1	0.6	6774.566	6253.126	6690.012
2	0.6	6766.972	6221.639	6699.352
3	0.6	6776.676	6194.275	6698.833
4	0.6	6800.024	6200.25	6704.684
5	0.6	6801.586	6193.047	6711.315
6	0.6	6772.027	6198.857	6717.874
7	0.6	6793.324	6208.962	6699.301
8	0.6	6764.746	6192.598	6709.063
9	0.6	6778.156	6209.789	6726.363
10	0.6	6790.819	6193.382	6706.925
1	0.7	6805.597	6308.878	6716.92
2	0.7	6870.746	6268.405	6754.942
3	0.7	6856.434	6258.2	6751.493
4	0.7	6886.547	6244.51	6756.641
5	0.7	6867.582	6242.204	6768.224
6	0.7	6866.767	6251.354	6768.112
7	0.7	6872.517	6237.552	6766.597
8	0.7	6842.732	6250.054	6757.329
9	0.7	6855.253	6235.387	6761.453
10	0.7	6851.159	6248.085	6756.129
1	0.8	6847.836	6367.928	6756.107
2	0.8	6916.439	6321.454	6781.651
3	0.8	6902.682	6306.809	6795.312
4	0.8	6916.867	6302.499	6800.307
5	0.8	6922.286	6291.34	6807.88
6	0.8	6971.94	6303.245	6794.646
7	0.8	6957.419	6273.731	6807.865
8	0.8	6942.138	6298.293	6801.92
9	0.8	6926.607	6296.041	6795.085
10	0.8	6950.409	6301.26	6812.843
1	0.9	6817.974	6417.932	6775.003
2	0.9	6864.049	6388.668	6801.541
3	0.9	6903.975	6375.768	6807.939
4	0.9	6942.761	6374.748	6828.841
5	0.9	6994.533	6359.959	6836.528
6	0.9	6982.07	6348.466	6847.772
7	0.9	6975.73	6351.51	6823.406
8	0.9	7019.292	6357.479	6849.442
9	0.9	7068.835	6349.178	6835.515
10	0.9	7009.447	6346.647	6840.79
1	1	6863.658	6485.376	6795.435
2	1	6855.906	6477.979	6812.688
3	1	6856.123	6479.946	6802.894
4	1	6855.509	6487.636	6804.984
5	1	6855.509	6487.636	6805.219
6	1	6855.509	6487.636	6805.219
7	1	6855.509	6487.636	6805.219
8	1	6855.509	6487.636	6804.984
9	1	6855.509	6487.636	6804.984
10	1	6855.509	6487.636	6804.984

Table S6. Model 5: Grid search (DIC) to determine the optimal alpha and beta of demonstrators. Various combinations of alpha, ranging from 0.1 to 1 with increments of 0.1, and beta, from 1 to 10 with increments of 1, were explored to identify the optimal parameter combination for model performance. The findings indicated that the best model fit is achieved with an alpha value of 0.2 and a beta value of 9.

		Beta										
		-	1	2	3	4	5	6	7	8	9	10
Alpha	0.1		6521.26	6341.26	6240.22	6173.44	6149.63	6139.62	6101.82	6113.72	6114.93	6121.37
	0.2		6344.98	6172.84	6133.27	6093.83	6105.8	6090.23	6094.05	6089.4	6086.84	6087.75
	0.3		6247.84	6132.32	6109.63	6109.02	6098.28	6091.75	6099.48	6099.28	6094.74	6100.17
	0.4		6220.1	6130.74	6128.3	6133.92	6121.55	6114.03	6140.6	6126.19	6120.56	6118.37
	0.5		6239.26	6178.57	6162.72	6164.48	6144.33	6165.72	6160.01	6152.88	6145.68	6160.57
	0.6		6253.13	6221.64	6194.27	6200.25	6193.05	6198.86	6208.96	6192.6	6209.79	6193.38
	0.7		6308.88	6268.4	6258.2	6244.51	6242.2	6251.35	6237.55	6250.05	6235.39	6248.08
	0.8		6367.93	6321.45	6306.81	6302.5	6291.34	6303.24	6273.73	6298.29	6296.04	6301.26
	0.9		6417.93	6388.67	6375.77	6374.75	6359.96	6348.47	6351.51	6357.48	6349.18	6346.65
	1.0		6485.38	6477.98	6479.95	6487.64	6487.64	6487.64	6487.64	6487.64	6487.64	6487.64

We have also changed the manuscript as follows.

On p 28

“We performed a grid search to systematically test different combinations of demonstrator learning rates ($\alpha_{demonstrator}$, range 0.1-1) and choice temperature ($\beta_{demonstrator}$, range 1 to 10). Based on the best fitting parameter values (Tables S5 and S6), we fixed $\alpha_{demonstrator}$ to 0.2 and $\beta_{demonstrator}$ to 9 for subsequent model estimations.”

Reviewer comment: The posterior distribution of the model parameters are also partially troublesome. The peaking of the distributions for alpha, beta, and rho at the end of the pre-specified parameter ranges suggests that given the current model of self choice and demonstrator predictability the sampler would have drawn sample beyond the pre-determined range. This implies that there might a model misspecification going on here. Alternatively, there might be a strong correlation between learning rate and temperature at play, which is frequently observed with Rescorla-Wagner type learning models: given a certain experimental variability, the model can either capture this with a very high learning rate and a low temperature or the other way around. A careful discussion and interpretation of these posterior distribution is lacking in the paper.

Response: Thank you for raising these points. The previous posterior distributions of alpha, beta and rho indeed peaked around the end of the bounded parameter ranges. In the process of increasing parameter ranges, we assessed the possibility of model misspecification in more

detail. Specifically, we tested variants of the previous models in which the stimulation parameter τ_{choice} was set as divisive instead of multiplicative with respect to v , thereby preserving the role of τ_{choice} (decreasing v). This change grants τ_{choice} a broader range, simultaneously eliminating any theoretical concerns about learning rates outside of the range 0-1.

The modification of Model 5 (as well as Model 4 and 6) was implemented in Equation 7,

$$\text{Previous Model} \quad AcP_{(i)} = AcP_{(i-1)} + \tau_{choice} * v * (predictability_i - AcP_{(i-1)})$$

$$\text{New Model} \quad AcP_{(i)} = AcP_{(i-1)} + (v/\tau_{choice}) * (predictability_i - AcP_{(i-1)})$$

And also, in Model 1-3,

$$Eq3 - D. Q(a)_i^{outcome} = Q(a)_i^{outcome} + \frac{\alpha}{\tau} (R_i^{demonstrator} - Q(a)_i^{outcome})$$

$$Eq4 - D. Q(a)_i^{action} = Q(a)_i^{action} + \frac{\kappa}{\tau} * (A_i^{demonstrator} - Q(a)_i^{action})$$

Note that we have reperformed all the model-related analyses based on this model.

With respect to the posterior distributions of parameters, the learning rate alpha is traditionally set to range from 0 to 1 to restrict the resulting values to the same range. Some models might allow reverse learning (alpha < 0) depending on the experimental design and the purpose of the studies; however, reverse learning would not make sense in our experiment. We therefore kept alpha bounded between 0 and 1.

With respect to beta, the reviewer pointed out that the posterior distribution peaked at the end of the pre-determined bound. We therefore extended the range to [0, 30] and have re-estimated the models, allowing a broader range of betas. The model comparison results stayed essentially the same and the posterior distributions no longer peak at the upper bound. Similarly, we also have changed the range of rho to [0, 3]. Again, the model comparison results remained the same and the distributions do not peak at the upper bound (Figures below). This alleviates any concern about model misspecification.

As the reviewer mentioned, alpha and beta often correlate due to the model specification or shared cognitive processes (Krefeld-Schwab et al., 2022). However, further correlation analyses show that this was not the case in our study. Specifically, correlation analyses between β and α as well as between β and v (learning rate in the best model) revealed no significant correlations (*choice for self*: β and α $r = .252$, $p = .180$; β and v $r = -.167$, $p = .377$; *prediction of demonstrator's actions*: β and α $r = -.033$, $p = .861$; β and v $r = -.253$, $p = .178$). These data provide little indication that model estimation in our study might have been affected by correlations between learning rate and inverse temperature. We now mention this in the revised manuscript on page 35:

In sum, with the new ranges of parameters, the best model remained the same (Model 5) in both *choice for self* and *prediction of demonstrator actions*. We have updated the model comparison results, the relevant figures, and the correlation between the stimulation parameters and the behavior. Also, changing the range of parameters showed that the distributions of beta and rho

do not peak at the boundary of the given range, thereby suggesting a sensible model specification. All the results and changes are shown below.

Figure S10. Posterior distributions of the parameters of Model 5. A) Posterior distributions in Model 5 in choice for self. B) Posterior distributions of the parameters in Model 5 in prediction of demonstrator actions.

A

B

Model 5 - Prediction

We also edited the manuscript as follows:

On p 30

“ τ ranged between 0 to 10, allowing for both stimulation-induced impairments and improvements of observational learning.”

On p 34

“transformed the range of beta, rho, tau parameters to [0-30], [0 – 3], [0-10] at the individual level.”

Figure 3. B) Model comparison for choice for self. Lower DIC values indicate better model fit. The model with predictability learning controlling action-based learning (Model 5) explained choice for self best.

Figure S5. Model comparison for prediction of demonstrator action. Lower DIC values indicate better model fit. The model where DMPFC/preSMA downregulation affected predictability learning in the Action-Only condition explained prediction of demonstrator actions best.

On p 14,

“In addition to model comparison, we cross-checked whether the model captured the behavioral pattern we found in model-free analyses. First, we tested if the stimulation parameter τ_{choice} indeed captured the behavioral difference between the DMPFC/preSMA and vertex condition. A one-sample *t*-test showed that the stimulation parameter τ_{choice} was significantly larger than 1, $t(29)=5.83, p < 0.001$, indicating a significant DMPFC/preSMA cTBS effect. Second, we assessed whether the stimulation parameter was related (across participants) with the stimulation-induced

behavioral change. This was confirmed by a significant correlation (Pearson's, $r = .34$, $p = 0.03$, one-tailed; Figure 4A) between the TMS-induced change in predictability learning rate and the performance difference between vertex and DMPFC/preSMA stimulation, in the Action-Only condition. Thus, the stimulation parameter τ_{choice} in the Predictability learning-Action model captured the stimulation-related variation in behavior as intended. By contrast, we did not find such a correlation ($r = .024$, $p = .45$, one-tailed, Figure 4B) for the Action learning model (Model 2) that examined a decrease of action-based learning without incorporating predictability. This suggests that the Predictability learning-Action model provides an adequate account of the mechanisms underlying the behavioral effects induced by TMS."

Figure 4. Relation between stimulation effects in model-based and model-free analyses. A) The difference in predictability learning rates in vertex and DMPFC/preSMA cTBS conditions positively related to the difference in the proportion of choosing a high-reward probability option between vertex and DMPFC/preSMA cTBS ($r = 0.340$, $p = 0.033$, one-tailed) for Model 5 (Predictability learning- Action). In contrast, B) there was little relation for the equivalent analysis with Model 2 (Action learning, $r = .024$, $p = .452$, one-tailed).

Moreover, the stimulation parameter τ_{choice} also showed a negative correlation with the proportion of high reward probability option choice in the DMPFC/preSMA condition, Pearson's $r = -0.52$, $p = 0.020$, one-tailed (Figure S7A). In contrast, the predictability learning rate in the vertex condition was not correlated with performance (Figure S7C), Pearson's $r = -.184$, $p = .330$. Thus, our results support the conclusion that the stimulation parameter in the model successfully captured the behavioral effects of stimulation."

on p 16,

"Similar to choice for self, we ascertained for the Action-Only condition with the superb demonstrator that the stimulation parameter $\tau_{prediction}$ of the best model accounted for the cTBS effect on the proportion of correct predictions of demonstrator actions. The corresponding correlation was positive, Pearson's $r = .354$, $p = 0.026$, one-tailed (Figure S7D). In the same condition, we also assessed the correlation between the difference of predictability learning rates between the vertex and the DMPFC/preSMA cTBS and the difference in the proportion of correct predictions of demonstrator actions. The correlation was also positive, but not statistically significant, Pearson's $r = 0.09$, $p = 0.31$, one-tailed (Figure S7C). This finding is compatible with

the notion that the process disrupted by DMPFC/preSMA stimulation is expressed in both choices and predictions.”

Figure S7. Correlations between model parameters and choice for self (A-B) or prediction of demonstrator actions (C and D). (A) The stimulation parameter T_{choice} (from the Predictability action model) correlated significantly with the proportion of choosing the higher reward probability options in the DMPFC/preSMA condition. (B) The equivalent analysis revealed no significant correlation for the Action Learning model (Model 2). (C) The difference in the proportion of correct predictions between DMPFC/preSMA cTBS and vertex was not significantly related to difference in learning rate between the vertex and DMPFC/preSMA cTBS conditions. (D) The stimulation parameter $T_{prediction}$ related positively to the proportion of correct predictions of demonstrator actions in the DMPFC/preSMA conditions.

Table S1. Parameter recovery. The simulated and recovered parameters of Model5 showed correlations greater than 0.8 for *choice for self* and above 0.6 for *prediction of demonstrator actions*.

	α	β	ν	τ	ρ	ω
Model5 Choice	0.94	0.94	0.95	0.93	0.99	0.84
Model5 Prediction	0.82	0.67	0.64	0.78	0.64	0.84

Figure S8. Confusion matrix of models for (A) choice for self and (B) prediction of demonstrator actions

A

		Fit model					
		Model 1	Model 2	Model 3	Model 4	Model 5	Model 6
Simulated model	Model 1	1	0	0	0	0	0
	Model 2	0.1	0.76	0.14	0	0	0
	Model 3	0.08	0.14	0.78	0	0	0
	Model 4	0	0.14	0	0.86	0	0
	Model 5	0.02	0	0.12	0	0.8	0.06
	Model 6	0	0	0	0.16	0.08	0.76

B

		Fit model					
		Model 1	Model 2	Model 3	Model 4	Model 5	Model 6
Simulated model	Model 1	0.24	0.18	0.06	0.24	0.22	0.06
	Model 2	0.16	0.28	0.02	0.16	0.36	0.02
	Model 3	0.22	0.12	0.04	0.22	0.34	0.06
	Model 4	0	0	0	1	0	0
	Model 5	0.02	0	0	0	0.96	0.02
	Model 6	0	0	0	0.04	0.04	0.92

Figure S9. Simulation of DMPFC/preSMA downregulation with the Predictability learning-Action model. A) Simulation of the Predictability learning-Action model with no TMS effect ($\tau = 1$). B) Simulation of the Predictability learning-Action model with DMPFC/preSMA downregulation ($\tau = 5$). With smaller τ , the DMPFC/preSMA condition in the Action-Only condition with the superb demonstrator showed a lower proportion of choosing the higher reward probability option compared to the vertex condition.

p 35,

“Additionally, learning rates and softmax temperature often correlate (Krefeld-Schwalb et al., 2022) which can limit interpretability of the findings of learning models. We found little evidence for this concern in our data. Specifically, correlation analyses between β and α as well as between β and u (learning rate in the best model) revealed no significant correlations (choice for self: β and α $r = .252$, $p = .180$; β and v $r = -.167$, $p = .377$; prediction of demonstrator’s actions: β and α $r = -.033$, $p = .861$; β and v $r = -.253$, $p = .178$).”

In a broad viewer of the modeling efforts, I still think that it offers very little beyond the model-free analysis of the mixed-models approach. The way the predictability model is constructed just replicates the experimental conditions for the demonstrator. Predictability (1-entropy) is just a measure of how much the choice history of the demonstrator deviates from 50/50 and this is exactly the how the demonstrators are constructed in the experiment. Because the neural manipulation is done off-line, the additional insight that a correlation of brain activity with model-derived variables does not apply for this study (i.e. not functional images of brain activity were collected).

Response: It is true that the model-free analysis shows interesting results by itself. However, we respectfully disagree with the statement that our modeling effort “offers very little beyond the model-free analysis of the mixed-models approach”. As detailed in the following, it would not be possible to draw any conclusions about the mechanisms affected by TMS just based on model-free analyses without computational modeling.

The DMPFC/preSMA cTBS effect was found exclusively in the superb demonstrator in the Action-Only condition, which targeted action-based learning. In contrast, the bad

demonstrator in the Action-Outcome condition mainly captured outcome-based learning. Based on the model-free analysis one would (correctly) conclude that DMPFC/preSMA plays a role in action learning. However, one would not know how exactly it achieves this role or (wrongly) assume that it achieves it through simple forms of learning from action prediction errors. Yet, our modeling explicitly tested and refuted this seemingly parsimonious explanation (Model 2) and linked the effect more specifically to predictability (Model 5). This would have been hard to do based on the model-free analyses, which did not reveal a simple stimulation effect in the superb demonstrator for the Action-Outcome condition. Thus, without any, or with less extensive, computational modeling, our paper would only show that the DMPFC/preSMA plays some causal role for action-based observational learning, but not provide any information on the specific learning mechanism DMPFC/preSMA implements.

Also, it is important to note that we have not included simply the static predictability of the demonstrator's behavior in the model, but a dynamic predictability updating process. By extension, disruption of DMPFC/preSMA function with cTBS interrupted predictability updating, not simply the prediction estimation process. We were able to reveal the causal role of the DMPFC/preSMA specifically for learning about the predictability of others' actions thanks to systematic assessment of two groups of models with a common structure (group 1: Models 1, 2, 3; group 2: Models 4, 5, 6). The two groups examined seemingly similar mechanisms at the group level but based on different underlying cognitive processes (Model 1 \approx Model 4, outcome-based learning disrupted, Model 2 \approx Model 5, action-based learning is disrupted, Model 3 \approx Model 6, both disrupted). Indeed, model comparisons clearly showed that our data were better fit by Model 5 than Model 2 (Δ DIC 273, which is substantially bigger than the minimal Δ DIC (5) considered meaningful by Spiegelhalter et al., 2002; 2003). Moreover, the fact that the stimulation parameters correlated with individual differences between vertex and DMPFC/preSMA cTBS conditions for Model 5 but not for the seemingly similar Model 2 highlights the distinct nature of the cognitive processes assumed by the two models. Revealing this difference was only possible through our computational models. Together, we could show that DMPFC/preSMA downregulation not simply causes decreased action-based learning but decreased action predictability learning, which in turn modulates action-based learning. Also note that the computational modeling results were drawn from all the social learning conditions, including the superb demonstrator in the Action-Outcome condition, allowing us to assess how the mixture of outcome-based and action-based learning comes about under DMPFC/preSMA downregulation. All of this could have not been achieved without computational modeling.

More generally, we note that predictability learning is a novel concept that can be linked to the previously established role of the DMPFC/preSMA in performance evaluation (Behrens et al., 2007, Ridderinkhof et al., 2004). Moreover, learning and using predictability to modulate action-based learning represents a form of meta-learning, which advances research on metacognition/meta-learning (Jiang et al., 2022, Silvetti et al., 2023). Together, our study and the model thus provide a more unified framework that comprehensively explains the multifaceted function of the DMPFC/preSMA.

We have updated the manuscript to highlight the findings as follows.

p 17,

“To our knowledge, this is the first study to show a causal role for DMPFC/preSMA in observational learning and to reveal that this role comprises specifically weighting the observational action-based learning rate with the predictability of others' actions. We were able

to dissociate these two possibilities thanks to systematic assessment of two groups of models with a common structure (group 1: Models 1, 2, 3; group 2: Models 4, 5, 6) that predicted similar behavioral patterns at the group level (Model 1 ≈ Model 4, Model 2 ≈ Model 5, Model 3 ≈ Model 6), but based on cTBS effects on different underlying cognitive processes. The distinguishability of Models 2 and 5 was corroborated by the fact that the stimulation parameters from Model 5 but not those of the seemingly similar Model 2 correlated with individual differences between vertex and DMPFC/preSMA cTBS conditions. Thereby, we could show that DMPFC/preSMA downregulation not simply causes decreased action-based learning but decreased action predictability learning, which in turn modulates action-based learning. We establish this not just at the group level, but also by showing that performance correlates across subjects with the individual TMS effects on specific model parameters.

We also note that action predictability learning is a novel concept that can be linked to the role of the DMPFC in performance evaluation (Behrens et al., 2007, Ridderinkhof et al., 2004). DMPFC has been functionally associated with monitoring and adjusting performance in response to behavioral errors of the agent themselves (Botvinick et al., 2001a; Holroyd et al., 2004; Ridderinkhof et al., 2004; Seo et al., 2014), observed errors of others (Yoshida et al., 2011) and irrelevant information evaluating one's ability (Mahmoodi et al., 2023). In keeping with a previous suggestion (Charpentier et al., 2020), we propose that action predictability provides a formal measure of demonstrator performance. The predictability of the demonstrator should be particularly relevant and diagnostic for demonstrator quality when only actions are observable (Najar et al., 2020). Our findings suggest that in such situations, the role of the DMPFC in using and optimizing observed information is relatively specific. The best fitting model indicates that learning about the predictability of demonstrator actions enables subsequent performance adjustment by dynamically setting the learning rate when weighting observed action prediction errors. Thus, our findings provide novel and causal evidence for the role of DMPFC in performance monitoring and adjustment during social learning.”

Reviewer #4 (Remarks to the Author):

Follow-up comment on the 1st comment by Reviewer #2:

Given the inherent difficulty of precisely specifying anatomical labels of the human brain normalized to a template (e.g., MNI and Talairach atlases), it is fair to describe the stimulation site as DMPFC/preSMA.

Response: Thank you very much for the comment.

Follow-up comment on the last comment by Reviewer #2:

The description of the computational models could be significantly improved.

For example, it is difficult to understand because the intuitive explanation of the formula, the description of the symbols within it, and the formula itself are described in separate locations. Moreover, the description of the Bayesian hierarchical model is unclear. In a Bayesian hierarchical model, each subject's individual-level parameter (e.g., learning rate) is assumed to be generated from the group-level distribution, such as a normal or beta distribution (its parameters are denoted as population-level parameters).

See <https://www.princeton.edu/~ndaw/d10.pdf>. As far as I understand, the authors assume the individual-level parameter, α , is drawn from the group-level Beta distribution that has two parameters (say μ and ν). Then, what does the description

“ $a \sim \text{dunif}(0.001, 0.999)$ ” mean? What is shown in Fig. A9A? The authors should provide a formal and comprehensive description.

Response: Thank you for raising these points. We will address each point separately.

1) Model description

We apologize that the description of the models was not intuitively understandable. We aimed to minimize redundancy by not repeating common elements in the models, which unfortunately may have compromised the connection to the equations for each model. To explain the models without repetition, we have now implemented color coding in Figure 3A to enhance the presentation of the model through diagrams. Also, we have introduced an extra figure to elucidate the relationships between the equations within each model, thereby improving the overall comprehensibility of the descriptions.

Figure 3. Models and model comparison for choice for self in the decision phase. A) A schematic of the compared models.

Figure S6. Schematic of the compared models including the equations involved in each model.

Next, in order to improve the comprehensibility of the symbols, we have changed the names of variables in the models, so that readers can easily associate them with those of a standard Q learning model, $Q(a)_i \rightarrow Q(a)_i^{outcome}$, $AP(a)_i \rightarrow Q(a)_i^{action}$. We have added also a table summarizing the meaning of all the symbols in the model.

Table S4. Description of the parameters in computational models

Parameters	Description	
α	Outcome-based learning rate	
β	Softmax inverse temperature	
κ	Action-based learning rate	
w	Parameter modulating the weight of outcome- versus action-based learning in Action-Outcome trials	
ρ	Perseverance parameter	
v	Learning rate for predictability of demonstrator actions	Only in Model 4,5,6
τ	Parameter capturing the DMPFC/preSMA cTBS effect	

- 2) We apologize for insufficient clarity in the description of the hierarchical modeling approach. We have also included more detail by adding a separate paragraph for the parameter estimation and model validation procedure.

On p34,

“Parameter estimation, model comparison and validity checks

*We used a hierarchical Bayesian approach for model estimation. The hierarchical procedure assumes that the model parameters of individual participants are drawn from a group-level distribution (Steingroever et al., 2013). This allows more reliable individual-level model estimation than separate estimation for each participant, as the individual-level parameter estimates are constrained by the group distribution. Group-level distributions for all model parameters were assumed to be beta distributions. The group-level means were assigned a uniform hyperprior on the interval [0.001, 0.999] and the group-level precisions were assigned a uniform hyperprior in the range [0.001, 10]. Beta distributions are typically defined by two shape parameters (a, b). We parameterized $a = \text{group level mean} * \text{group-level precision}$, $b = (1 - \text{group level mean}) * \text{group-level precision}$ (Hill et al., 2017) and transformed the range of beta, rho, tau parameters to [0-30], [0–3], [0-10] at the individual-level. We inferred posterior distributions for all model parameters using Markov chain Monte Carlo (MCMC) sampling, as implemented in the JAGS package via the R2jags interface (Su & Yajima, 2012). We ran three independent MCMC chains with different starting values per model parameter and collected 40,000 posterior samples per chain. We discarded the first 10,000 iterations of each chain as burn-in. In addition, we only used every 5th iteration to remove autocorrelation. Consequently, we obtained 18,000 representative samples per parameter per model. We used the Deviance Information Criterion (DIC) for the model comparisons (Spiegelhalter et al., 2002). The DIC is an index of the goodness of model fit, penalized by its effective number of parameters. A smaller DIC indicates a better model fit. The posterior distributions of the parameters are shown in Figure S10.”*

References

- Behrens, T. E. J., Woolrich, M. W., Walton, M. E., & Rushworth, M. F. S. (2007). Learning the value of information in an uncertain world. *Nature Neuroscience*, 10(9), 1214–1221. <https://doi.org/10.1038/nn1954>
- Botvinick, M. M., Braver, T. S., Barch, D. M., Carter, C. S., & Cohen, J. D. (2001). Conflict monitoring and cognitive control. *Psychological Review*, 108(3), 624–652. <https://doi.org/10.1037/0033-295X.108.3.624>
- Charpentier, C. J., Iigaya, K., & O’Doherty, J. P. (2020). A Neuro-computational Account of Arbitration between Choice Imitation and Goal Emulation during Human Observational Learning. *Neuron*, 106(4), 687-699.e7. <https://doi.org/10.1016/j.neuron.2020.02.028>
- Hill, C. A., Suzuki, S., Polania, R., Moisa, M., O’Doherty, J. P., & Ruff, C. C. (2017). A causal account of the brain network computations underlying strategic social behavior. *Nature Neuroscience*, 20(8), 1142–1149. <https://doi.org/10.1038/nn.4602>
- Mahmoodi, A., Luo, S., Harbison, C., Piray, P., & Rushworth, M. (2023). Human hippocampus and dorsomedial prefrontal cortex infer and update latent causes during social interaction. *bioRxiv*, 2023-09.

- Najar, A., Bonnet, E., Bahrami, B., & Palminteri, S. (2020). The actions of others act as a pseudo-reward to drive imitation in the context of social reinforcement learning. *PLoS Biology*, 18(12), 1–25. <https://doi.org/10.1371/journal.pbio.3001028>
- Ridderinkhof, K. R., Ullsperger, M., Crone, E. A., & Nieuwenhuis, S. (2004). The role of the medial frontal cortex in cognitive control. *Science*, 306(5695), 443–447. <https://doi.org/10.1126/science.1100301>
- Spiegelhalter, D. J., Best, N. G., Carlin, B. P., & Van Der Linde, A. (2002). Bayesian measures of model complexity and fit. *Journal of the Royal Statistical Society Series B: Statistical Methodology*, 64(4), 583–639.
- Spiegelhalter, D., Thomas, A., Best, N., & Lunn, D. (2003). WinBUGS user manual.
- Steingroever, H., Wetzels, R., Horstmann, A., Neumann, J., & Wagenmakers, E.-J. (2013). Performance of healthy participants on the Iowa Gambling Task. *Psychological Assessment*, 25(1), 180.
- Su, Y.-S., & Yajima, M. (2012). *R2jags: a package for running jags from R. R package version 0.03-08*.
- Sutton, R. S., & Barto, A. G. (2018). Reinforcement learning: An introduction. MIT press.
- Yoshida, K., Saito, N., Iriki, A., & Isoda, M. (2011). Representation of Others' Action by Neurons in Monkey Medial Frontal Cortex. *Current Biology*, 21(3), 249–253. <https://doi.org/10.1016/j.cub.2011.01.004>

REVIEWER COMMENTS

Reviewer #3 (Remarks to the Author):

I very much appreciate the effort and diligence that the authors took in addressing my comments. The additional analyses and improvements to the models and the model fitting really convinced me of the results and have improved the impact of the study. Furthermore, also liked the detailed arguments and interpretation of the differences between the model families and the models themselves and how that crystallizes the effect of cTBS. Congratulations on a well-written paper and a well-argued interpretation. I support publication in Nature Communications now.

Reviewer #4 (Remarks to the Author):

I would like to thank the authors for their thorough efforts to revise the manuscript. The structure of the model is now clearly presented, which I greatly appreciate.

By reading the description of the computational modeling, unfortunately, I have found a critical flaw in the model fitting. The confusion matrix for the prediction of demonstrator actions (Fig. S9b) suggests that the model comparison is not effective, contrary to the authors' claim that "the models were identifiable." For instance, the matrix indicates that Model 5 is identified as the best model with a probability of 0.34, even when the data is simulated by a simpler model (i.e., Model 3), which exemplifies the identification problem. Furthermore, the probability that the true model (i.e., Model 3) is identified as the best-fit model is only 0.04. The observed pattern of non-identifiability may explain the results of the model comparison presented in Fig. S5. Specifically, it suggests the possibility that Model 5 provided the best prediction, despite the true generative model being simpler.

REVIEWER COMMENTS

Reviewer #3 (Remarks to the Author):

I very much appreciate the effort and diligence that the authors took in addressing my comments. The additional analyses and improvements to the models and the model fitting really convinced me of the results and have improved the impact of the study. Furthermore, I also liked the detailed arguments and interpretation of the differences between the model families and the models themselves and how that crystallizes the effect of cTBS. Congratulations on a well-written paper and a well-argued interpretation. I support publication in Nature Communications now.

Response: We sincerely appreciate your comments and thank you for the helpful questions that have significantly improved our manuscript.

Reviewer #4 (Remarks to the Author):

I would like to thank the authors for their thorough efforts to revise the manuscript. The structure of the model is now clearly presented, which I greatly appreciate.

By reading the description of the computational modeling, unfortunately, I have found a critical flaw in the model fitting. The confusion matrix for the prediction of demonstrator actions (Fig. S9b) suggests that the model comparison is not effective, contrary to the authors' claim that "the models were identifiable." For instance, the matrix indicates that Model 5 is identified as the best model with a probability of 0.34, even when the data is simulated by a simpler model (i.e., Model 3), which exemplifies the identification problem. Furthermore, the probability that the true model (i.e., Model 3) is identified as the best-fit model is only 0.04. The observed pattern of non-identifiability may explain the results of the model comparison presented in Fig. S5. Specifically, it suggests the possibility that Model 5 provided the best prediction, despite the true generative model being simpler.

Response: Thank you for your comment. We understand the reviewer's concern and agree that the results of one model recovery analysis in the submitted manuscript may give this impression. However, we want to point out that the model recovery of the *prediction of demonstrator actions* is not actually "a critical flaw in the model fitting".

The model recovery analysis in Figure S9A (*choice for self*) clearly showed that the models are identifiable. Since the models for *prediction of demonstrator actions* are identical to the ones used for *choice for self*, they are clearly identifiable as well if the same parameter settings are applied. This is because model recovery analyses are independent of actual data, and only test whether simulated data from models are accurately recovered by the same model and not others. This process can obviously be sensitive to the input parameter ranges even for identical models – in other words, models that are successfully recovered in simulations with certain parameter ranges may not be recovered well in another setting where different ranges of parameters are used.

In our model recovery analyses, we used parameter ranges found empirically, i.e., the distributions of the parameters obtained from model fitting. This would be reasonable in many

cases, but we missed that the distribution of some parameters for the *prediction of demonstrator actions* were very narrow (see Figure S10B). The resulting narrow parameter range we used for the simulations of the *prediction of demonstrator actions* thus made the model recovery difficult, as opposed to the wider range we used for the recovery analysis of the choices for self (which clearly shows that the models are recoverable if the parameters are allowed to vary; see the difference between the Figure S10A and S10B). The model recovery for the *prediction of demonstrator actions* therefore is somewhat misleading and even unnecessary, given that we do not need to decide which model is best for *prediction of demonstrator actions*. We use the same model as for *choice for self* anyway, to allow comparison of TMS effects, regardless of what the model recovery shows. However, the editor (Dr. Montijn) advised us to keep it in and provide additional explanations. Thus, we revised the manuscript to explain that the non-identifiable model recovery results for *prediction of demonstrator actions* reflects the narrow ranges of the fitted parameters and that the only model recovery that matters for the conclusions is that for *choice for self*.

On p35,

For the simulation, model parameters were chosen from the distribution of fitted parameters. After simulation, we fitted models on the data generated by each model and report the confusion matrix (Figure S9). The model recovery confirmed that the models were identifiable for *choice for self* but there was a bit weaker distinction for models for *prediction of demonstrator actions*. This may be due to the narrow range of fitted parameters as shown in Figure S10B. However, as we applied the same models for both behaviors, the model recovery results for *choice for self* provide evidence that these models are clearly identifiable in principle. Importantly, the model selection was entirely based on *choice for self* and the corresponding model recovery is the only one that matters for the conclusions of the study.

REVIEWERS' COMMENTS

Reviewer #4 (Remarks to the Author):

I acknowledge the authors for their responses to my comments. Overall, I believe the conceptual advance and novelty of this study lie in the model-free analyses, which elucidate the functional role of the dmPFC in observational learning. In other words, the significance of this study is not undermined by the shortcomings of model fitting.

Having said that, I would like to comment on their model recovery analysis. It is reasonable that the results of the recovery analysis depend on the range of simulation parameters (<https://elifesciences.org/articles/49547>). I would suggest that the authors transparently discuss that model identifiability in this study should be carefully evaluated (i.e., the models are not identifiable when using the parameter ranges found empirically) in the Discussion section as a limitation, so that readers can evaluate the validity of the model fitting findings.

Reviewer #4 (Remarks to the Author):

I acknowledge the authors for their responses to my comments. Overall, I believe the conceptual advance and novelty of this study lie in the model-free analyses, which elucidate the functional role of the dmPFC in observational learning. In other words, the significance of this study is not undermined by the shortcomings of model fitting.

Having said that, I would like to comment on their model recovery analysis. It is reasonable that the results of the recovery analysis depend on the range of simulation parameters (<https://elifesciences.org/articles/49547>). I would suggest that the authors transparently discuss that model identifiability in this study should be carefully evaluated (i.e., the models are not identifiable when using the parameter ranges found empirically) in the Discussion section as a limitation, so that readers can evaluate the validity of the model fitting findings.

Response: Thank you very much for the comment. Now we have included a paragraph discussing about model identifiability in this study in Discussion section as below.

On p18,

“The model recovery confirmed that the models were identifiable for choice for self but there was a bit weaker distinction for models for prediction of demonstrator actions. This may be due to the difference between the range of fitted parameters for the models for choice for self and prediction of demonstrator actions (Wilson & Collins, 2019). However, as we applied the same models for both behaviors, the model recovery results for choice for self provide evidence that these models are clearly identifiable in principle. Importantly, the model selection was entirely based on choice for self and the corresponding model recovery is the only one that matters for the conclusions of the study.”